# A gene expression signature of TREM2[hi] macrophages and γδ T cells predicts immunotherapy response

Donghai Xiong[1], Yian Wang[1] & Ming You[1✉]

Identifying factors underlying resistance to immune checkpoint therapy (ICT) is still challenging. Most cancer patients do not respond to ICT and the availability of the predictive biomarkers is limited. Here, we re-analyze a publicly available single-cell RNA sequencing (scRNA-seq) dataset of melanoma samples of patients subjected to ICT and identify a subset of macrophages overexpressing TREM2 and a subset of gammadelta T cells that are both overrepresented in the non-responding tumors. In addition, the percentage of a B cell subset is significantly lower in the non-responders. The presence of these immune cell subtypes is corroborated in other publicly available scRNA-seq datasets. The analyses of bulk RNA-seq datasets of the melanoma samples identify and validate a signature - ImmuneCells.Sig - enriched with the genes characteristic of the above immune cell subsets to predict response to immunotherapy. ImmuneCells.Sig could represent a valuable tool for clinical decision making in patients receiving immunotherapy.

[1] Center for Disease Prevention Research and Department of Pharmacology and Toxicology, Medical College of Wisconsin, Milwaukee, WI 53226, USA.
✉email: myou@mcw.edu

While immune checkpoint therapies (ICT) have improved outcomes for some cancer patients, most patients do not respond to ICT. Previous whole-exome sequencing (WES) and transcriptome sequencing of tumors identified multiple factors that are associated with favorable ICT outcome, including expression of PD-L1[1], high tumor mutational burden[2], and the presence of tumor-infiltrating CD8[+] T cells[3]. Markers indicative of unfavorable response include defects in IFNγ pathways or antigen presentation[4,5]. While these studies represented a first step in identifying biomarkers, studies using single-cell RNA sequencing (scRNA-seq) have the potential to greatly improve the identification of factors underlying ICT outcomes. For example, one scRNA-seq study of 48 tumor biopsies of responding and non-responding tumors after ICT treatment has the potential to be insightful given the number of patients and high quality data[6].

To determine if some types of immune cells and their subclusters are associated with ICT outcomes, we analyze the scRNA-seq datasets from multiple outstanding studies[6–8] and identify the immune cell subpopulations that could play an important role in determining ICT responsiveness. The analysis of several additional bulk RNA-seq datasets of melanoma[9–12] identifies and validates an ICT outcome signature -ImmuneCells. Sig - enriched with the genes characteristic of the immune cell subsets detected in the scRNA-seq studies. It predicts the ICT outcomes of melanoma patients more accurately than the previously reported ICT response signatures.

Specifically, we find that a subset of macrophages (cluster 12) and a subset of gammadelta (γδ) T cells (cluster 21) are highly enriched in the ICT non-responding tumors. On the other hand, the percentage of a subset of B cells (cluster 22) is significantly smaller in the ICT non-responders compared to the responders. The validated ImmuneCells.Sig ICT outcome signature is enriched with the genes characteristic of the above three immune cell subsets. It can predict the ICT outcomes of melanoma patients more accurately than the previous outstanding signatures, thereby supporting the role of these specific types of immune cells in affecting the ICT outcomes. These findings substantially extend our understanding of the factors associated with ICT responsiveness. Our results may warrant further investigation in the cancer immunotherapy setting.

## Results

**Association of immune cell populations with ICT outcome.** We utilized the Seurat package[13,14] to perform fine clustering of the original 16,291 single cells based on raw data from a previous melanoma study[6]. The melanoma patient response categories were defined by RECIST (Response evaluation criteria in solid tumors) as: complete response (CR) and partial response (PR) for responders, or stable disease (SD) and progressive disease (PD) for non-responders[15]. Progression-free survival was also considered in distinguishing the responders from non-responders. To relate molecular and cellular variables with responses of individual lesions to therapy, the previous study classified each of the 48 tumor samples based on radiologic assessments into progression/non-responder (NR; $n = 31$, including SD/PD samples) or regression/responder (R; $n = 17$, including CR/PR samples)[6]. The gene expression data of single cells from tumors with different ICT outcomes, i.e., regression/responder (Responder - 'R'; n.patients = 17; n.cells = 5564) and progression/non-responder (Non-Responder - 'NR'; n.patients = 31; n.cells = 10,727), were aligned and projected in two-dimensional space through uniform manifold approximation and projection (UMAP)[16] to allow the identification of ICT-outcome-associated immune cell populations. This analysis generated 23 cell clusters across all samples

(Fig. 1a). The percentages of immune cells from each cluster from responding and non-responding melanoma groups were calculated (Supplementary Table 1).

We utilized gene expression patterns of canonical markers to classify the 23 clusters into 10 major immune cell populations (Fig. 1b and Supplementary Fig. 1a): CD8[+] T cells (CD3[+]CD8A[+]CD4[−], clusters 1,4,5,7,10,11,20); CD4[+] T cells (CD3[+]CD8A[−]CD4[+], cluster 3); Regulatory T cells (Tregs) (FOXP3[+], cluster 2); MKI67hi Lymph. (MKI67[+], clusters 9,16); B cells (CD19[+], clusters 13,14,17,22); Plasma cells (MZB1[+], cluster 18); NK cells (NCR1[+]NCAM1[+], cluster 15); γδ T cells (i.e., Tgd cells, CD3[+]CD8A[−]CD4[−], clusters 8,21); Macrophages (MARCO[+]MERTK[+], clusters 6,12,23); and Dendritic cells (FCER1A[+], cluster 19). The identification of γδ T cells is further justified as follows. The NK cells in cluster 15 expressed the NK cell markers NCR1 and NCAM1, which were not expressed in γδ T cells in clusters 8 and 21 (Supplementary Fig. 1a). Also, the NK cells (cluster 15) do not express CD3 markers, whereas CD3 markers were expressed in the adjacent clusters (8 and 21) that were characterized as γδ T cells based on the combination CD3[+]CD4[−]CD8[−]. In addition, we validated our defined γδ T lymphocytes by the expression of the published gene expression signatures of γδ T cells[17], which requires scoring the following two gene sets: the positive gene set (CD3D, CD3E, TRDC, TRGC1, and TRGC2), and the negative gene set (CD8A and CD8B) for each single cell. Specifically, following this published approach, to identify γδ T lymphocytes exhaustively and without NK and T-cell CD8 false-positives, we utilized the established γδ signature that combines the above two gene sets that were scored for each single cell and visualized in the UMAP by Single-Cell Signature Explorer[18]. As shown in Supplementary Fig. 1b, the γδ signature scores were highest for clusters 8 and 21 but much lower in the other clusters. These data further support our assignment of γδ T lymphocytes to clusters 8 and 21.

We tested the 23 immune cell clusters for their percentage differences between the non-responders and responders at the patient level (Fig. 1c and Supplementary Fig. 2). The results were compared to the results of the integrative analysis to calculate the overall fold changes between the non-responder and responder groups. Some of these immune cell clusters differed quantitatively between ICT responders (R) and non-responders (NR), including the Clusters 6, 9, 12, 13, 14, 17, 19, 21, 22 (Fig. 1c), which was supported by the integrative analysis combining cells from all patients (Fig. 1d). Furthermore, using more than 6-fold differences as a biologically significant threshold[19], we identified three clusters (12, 21, and 22) that exceeded this criterion. Cluster 12 (a macrophage cluster) and cluster 21 (a γδ T-cell cluster) cells were 15.1-fold and 12.1-fold higher, respectively, in ICT non-responders versus responders (Fig. 1d and Supplementary Table 1). In contrast, the percentage of cluster 22 cells (a B-cell cluster) was 9.3-fold lower in the non-responders. Two other B-cell clusters (cluster 13 and 17) were 5.8- and 4.1-fold lower, respectively, in the non-responders. The remaining 18 clusters exhibited only minor (1.1- to 2.9-fold) differences between responders non-responders (Fig. 1d and Supplementary Table 1). Our approach is similar to the approach used in the previous scRNA-seq study of the effects of the immunotherapy on changing the percentages of different immune cell subpopulations[20]. They compared the percentage of cells in individual clusters for different conditions of control, anti-PD-1, anti-CTLA-4, and anti-PD-1/anti-CTLA-4. In this way, they identified a number of immune cell subclusters that could be associated with the variation of the efficacy of the cancer immunotherapy.

To account for clinical differences, we divided the melanoma samples into subgroups according to three factors: (1) ICT

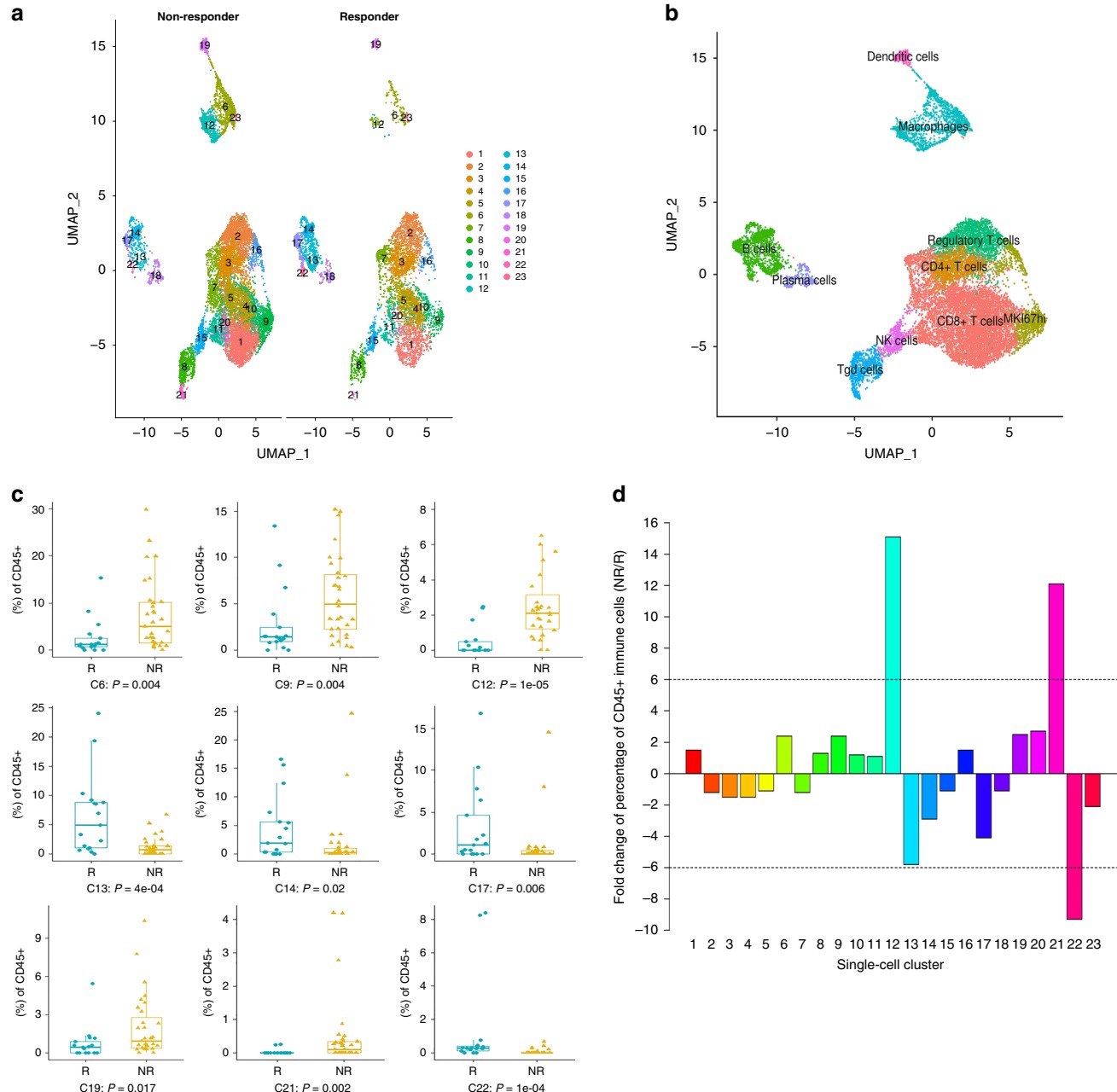

**Fig. 1 Identification of intratumoral immune cell populations by scRNA-seq. The scRNA-seq dataset - GSE120575 was analyzed. a** Uniform manifold approximation and projection (UMAP) plot of intratumoral immune cells that were classified into 23 clusters from the two groups of melanoma samples of distinct immune checkpoint therapy outcomes (NR [non-responder] group versus R [responder] group). **b** UMAP plot of the 10 major immune cell populations. **c** Comparison of the cell cluster percentage changes between the NR and R groups. Boxplots showed the results for the nine immune cell clusters with significant changes. Center line, median. Box limits, upper and lower quartiles. Whiskers, 1.5 interquartile range. Points beyond whiskers, outliers. The two-sided Wilcoxon tests were performed with no adjustment for multiple comparisons. **d** The fold changes of the percentages of each of the 23 single-cell clusters comparing the NR group to the R group.

outcomes, (2) sample collection time (before or after ICT), and (3) treatment schemes (Supplementary Table 2). There were only six groups with sufficient numbers of samples and cells to compare between non-responders and responders, i.e., G1 vs G7 ('NR-before-anti-PD-1' vs 'R-before-anti-PD-1'), G4 vs G10 ('NR-after-anti-PD-1' vs 'R-after-anti-PD-1'), and G6 vs G12 ('NR-after-anti-CTLA4 + PD-1' vs 'R-after-anti-CTLA4 + PD-1'). Stratified analyses showed similar results of cell cluster percentage changes as those in the integrative analysis (Fig. 1d and Supplementary Fig. 3).

**TREM2$^{hi}$ macrophages may contribute to ICT resistance.** Of the macrophage populations in clusters 6, 12, and 23 (Fig. 1a, b), differences between the R and NR groups were not significant for cluster 6 (2.4-fold higher in NR) and cluster 23 (2.1-fold lower in NR). However, cluster 12 macrophages were 15.1-fold higher in NR (4.88%) versus R (0.32%). This enrichment of cluster 12 in non-responders suggests that it may be associated with ICT resistance. Single-cell differential expression analyses were performed to assess the most characteristic gene expression differences in clusters 6, 12, and 23 (Fig. 2 and Supplementary Data 1,

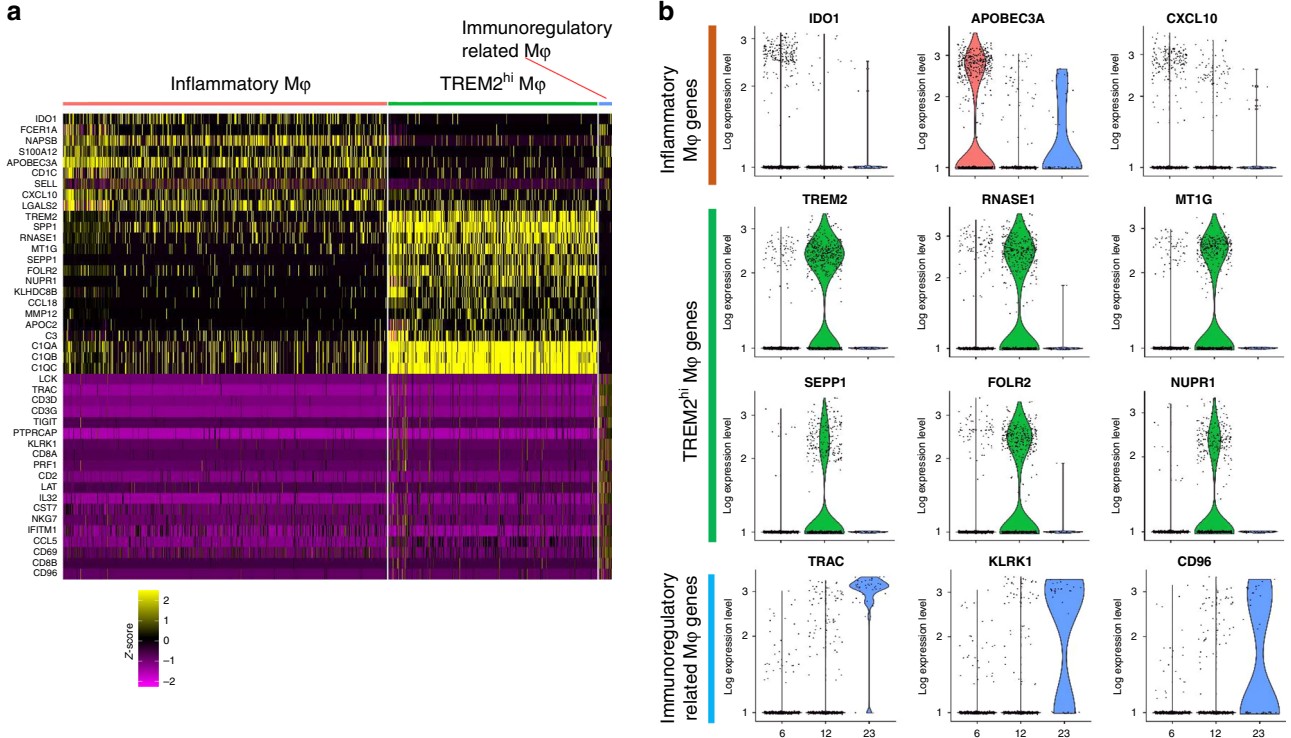

**Fig. 2 Subsets of macrophages in the melanoma tumors.** The scRNA-seq dataset - GSE120575 was used in this analysis. **a** Heatmap of z-scored expression of the top up-regulated genes of each macrophage subpopulation versus the other two macrophage subpopulations. **b** Violin plots of log-transformed gene expression of selected genes showing statistically significant upregulation in inflammatory macrophages (top), TREM2hi macrophages (center), and Immunoregulatory related macrophages (bottom).

2, and 3). Cluster 12 (35.9% of all macrophages, Supplementary Fig. 4) overexpressed *TREM2* (Fig. 2) so was named as TREM2hi Mφ (Mφ = macrophages). The TREM2hi Mφ that were enriched in non-responders displayed a unique signature with over-expression of *SPP1, RNASE1, MT1G, SEPP1, FOLR2, NUPR1, KLHDC8B, CCL18, MMP12,* and *APOC2* along with key complement system genes (*C3, C1QA, C1QB,* and *C1QC*) (Fig. 2). Cluster 6 cells (61.6% of all Mφ, Supplementary Fig. 4), over-expressed the immunosuppressive protein indoleamine 2,3-dioxygenase 1 (*IDO1*) (Fig. 2), as well as several inflammatory markers (*FCER1A, S100A12, APOBEC3A, SELL,* and *CXCL10*). Ingenuity Pathway Analysis (IPA) confirmed that inflammatory markers were significantly overexpressed in cluster 6 versus other macrophages (adjusted $P = 3.93E-10$, activation $Z$ score = 2.01, Supplementary Fig. 5; $P$ values throughout this paper are adjusted by using Bonferroni correction unless otherwise declared). Therefore, we named cluster 6 as Inflammatory Mφ. Cluster 23 cells (2.5% of all Mφ, Supplementary Fig. 4) were 2.1-fold higher in responders and expressed several genes involved in immune regulation, i.e., *LCK, TIGIT, PTPRCAP, KLRK1, LAT, IL32, IFITM1,* and *CCL5* (Fig. 2a)[21]. Cluster 23 was thus named as Immunoregulatory related Mφ.

**Significantly enriched pathways in TREM2hi macrophages.** To identify if functional heterogeneity of these macrophage subsets could be associated with ICT outcomes, we performed 'Reactome pathways' analysis for macrophages based on cluster-specific genes detected by Seurat (Supplementary Data 4, 5, and 6). Each macrophage subset was significantly enriched for specific molecular pathways. Inflammatory Mφ (cluster 6) were enriched for FCERI signaling and several FCERI-mediated pathways (NF-kappaB activation, Ca2+ mobilization and MAPK activation; Supplementary Fig. 6). The Immunoregulatory related Mφ

(cluster 23) were most significantly enriched for pathways involving Regulation of expression of SLITs and ROBOs and Signaling by ROBO receptors (Supplementary Fig. 6). TREM2hi Mφ (cluster 12), which showed the greatest percentage elevation in ICT non-responders, was enriched for multiple pathways underlying complement activation (complement cascade and its regulation, initial triggering of complement, creation of C4 and C2 activators, and classical antibody-mediated complement activation; Supplementary Fig. 6). These findings were consistent with overexpression of complement system genes in TREM2hi Mφ, including complement C1q chains (*C1QA, C1QB,* and *C1QC*), complement C2 and C3 (Supplementary Fig. 7a). These genes were either not expressed, or at very low levels in macrophage clusters 6 and 23. TREM2hi macrophages also over-expressed M2 polarization genes (*MMP14, CD276, FN1, MRC1, CCL13, CCL18, LYVE1, PDCD1LG2* (PD-L2), *MMP9, TGFB2,* and *ARG2*; Supplementary Fig. 7a). TREM2hi macrophages may therefore be functionally proximal to M2 polarization macrophages and could block the anti-tumor activities of ICT and contribute to ICT resistance.

**Validation of the TREM2hi macrophage signature.** Since TREM2hi macrophages correlated with ICT resistance, we determined if tumors enriched in TREM2hi macrophages were associated with poor ICT outcomes. Based on the overexpressed genes of this macrophage subset, we developed a 40-gene set to characterize TREM2hi macrophages, which included the genes highly correlated with TREM2 expression (those for the complement system or M2 polarization), and other overexpressed genes (Supplementary Fig. 7a). In order to test if this TREM2hi macrophage signature was correlated with ICT resistance, we analyzed two publicly available gene expression datasets of tumor samples from melanoma patients treated with

immunotherapy[9,10]. The GSVA scores of the TREM2[hi] macrophage geneset were significantly higher in the ICT non-responders than the responders (Supplementary Fig. 7b, c), suggesting that melanomas in non-responders were enriched for TREM2[hi] macrophages. The analyses of the GSVA scores of this 40-gene set verified the specificity of this gene set to characterize the TREM2[hi] macrophages among all groups of macrophages (Supplementary Fig. 7d).

**Association of γδ T- and B-cell subsets with ICT outcome**. We also identified two clusters of γδ T cells (927 cells total; clusters 8 and 21, Fig. 1, and Supplementary Table 1). The more common type of γδ T cells (cluster 8, $n = 781$) was not significantly different between non-responders and responders. However, a rare type of γδ T cells (cluster 21, $n = 146$) was 12.1-fold higher in the NR group (1.31% versus 0.11% in R; Fig. 1 and Supplementary Table 1). This fold-difference is the second largest of all 23 clusters. These findings suggest that the cluster 21 γδ T cells (named as Tgd_c21) may contribute to ICT resistance. Single-cell differential expression analyses compared Tgd_c21 to Tgd_c8 cells (Supplementary Data 7), with the top 20 marker genes shown in Fig. 3a. The top Tgd_c21 marker genes included *RRM2*, *BIRC5*, *SPC24*, *UBE2C*, and *CDCA5*. GSEA pathway analyses[22] revealed multiple pathway changes that could be correlated with the contribution of Tgd_c21 cells to ICT resistance, including significant reductions in ligand-receptor binding capacity, IFNα and IFNβ signaling, IFN-γ response, and immunoregulatory interactions (Fig. 3c). Oncogenic (HALLMARK_E2F_TARGETS) and cell cycle pathways were also activated in Tgd_c21 (Fig. 3c). Thus, Tgd_c21 cells may represent a previously unidentified class of γδ T cells that may impair anti-tumor immune functions.

We also identified a correlation between the presence of B cells and ICT response. All four B-cell clusters (13, 14, 17, and 22) were less abundant in the ICT non-responders, which suggests that tumor-associated B cells, in general, are associated with favorable ICT response. Most notably, the percentage of cluster 22 B cells (named as B_c22) was 9.3-fold lower in NR versus R (Fig. 1 and Supplementary Table 1); this is the largest deficit in NR tumors across all 23 clusters. We performed differential expression and pathway enrichment analyses comparing B_c22 to other B-cell clusters (Supplementary Data 8). The top 20 marker genes for each B-cell cluster were determined (Fig. 3b). GSEA pathway analysis showed that B_c22 cells had significantly reduced oncogenic signaling, including Toll receptor signaling/cascades, NOTCH1, MAPK, and MYC signaling pathways (Fig. 3c). The significant enrichment of B_c22 cells in ICT responders may therefore contribute to the attenuation of oncogenic signaling in the tumor microenvironment (TME) to enhance the anti-tumor effect in response to ICT.

**Validation in the other scRNA-seq datasets of ICT patients**. To validate the results we found based on the initial scRNA-seq data, we downloaded and re-analyzed another scRNA-seq dataset of melanoma with corresponding immunotherapy efficacy data[7]. This dataset did not have γδ T-cell data available. Interestingly, the deeper clustering of the macrophages and B cells sequenced by this study showed the existence of similar macrophage and B-cell subpopulations that resemble our identified TREM2[hi] macrophages and B_c22 B cells (Supplementary Fig. 8a, b). Specifically, the 'Mac_c1' macrophage subcluster overexpressed the TREM2[hi] macrophage marker genes such (*TREM2*, *SPP1*, *RNASE1*, *MT1G*, *SEPP1*, *FOLR2*, *KLHDC8B*, *CCL18*, *MMP12*, *APOC2*, *C3*, *C1QA*, *C1QB*, and *C1QC*; Supplementary Fig. 8c); the 'B_s1' B-cell subcluster overexpressed the B_c22 B cell marker genes (*ABCA6*, *LEF1*, *FGR*, *IL2RA*, *ITGAX*, and *IL7*;

(Supplementary Fig. 8d). More importantly, we validated the behavior of these two immune cell subpopulations in the context of the response to immunotherapy. We scored each cell based on its overall expression (OE) of the corresponding signature following the previous approach[7], i.e., scoring each Mac_c1 macrophage for its TREM2[hi] macrophage signature and each B_s1 B cell for its B_c22 B-cell signature, and compared these between the non-responder and control groups. In this dataset, the Mac_c1 macrophage subset had significantly higher overall expression of the TREM2[hi] macrophage signature in the immunotherapy non-responders than in the control samples (Supplementary Fig. 8e). The B_s1 B-cell subset had significantly lower overall expression of the B_c22 B-cell signature in the immunotherapy non-responders than in the control samples (Supplementary Fig. 8f). These results supported the findings in our initial scRNA-seq dataset of the changes in TREM2[hi] macrophages and B_c22 B cells in response to immunotherapy.

We also analyzed a single-cell RNA-seq dataset of basal cell carcinoma (BCC) patients before and after anti-PD-1 therapy[8]. We found that the results of our study can be generalized to BCC treated with ICT. Although this BCC scRNA-seq dataset did not sequence the γδ T cells, the results for macrophages and B cells in this BCC dataset are similar to our findings for the melanoma dataset. First, we did general clustering analyses and identified the overall macrophages and B cells populations (Supplementary Fig. 9a). Then we performed finer clustering and identified the macrophages and B-cell subpopulations from the BCC tumors that are similar to the TREM2[hi] macrophages and B_c22 B cells in the initial melanoma samples (Supplementary Fig. 9b–e). In the BCC dataset, the Mac_s2 macrophage subcluster overexpressed the TREM2[hi] macrophage marker genes (*TREM2*, *FOLR2*, *MMP12*, *C1QA*, *C1QB*, and *C1Qc*; Supplementary Fig. 9d); the B_sc2 B-cell subcluster overexpressed the B_c22 B cells marker genes (*TRAC*, *IL2RA*, *ITGB1*, *ZBTB32*, *TRAF1*, and *CCND2*; Supplementary Fig. 9e). As before, we validated the overall expression changes of the TREM2[hi] macrophage signature of the Mac_s2 macrophages and the B_c22 signature of the B_sc2 B cells in response to the anti-PD-1 immunotherapy in this BCC dataset[8]. Specifically, the Mac_s2 macrophage subset had significantly decreased overall expression of the TREM2[hi] macrophage signature in the responsive BCC tumors after anti-PD-1 therapy when compared to the pretreatment BCC samples (Supplementary Fig. 9f). The B_sc2 B-cell subset had significantly higher overall expression of the B_c22 signature in the post anti-PD-1 therapy in the responsive BCC tumors than in the pretreatment BCC samples (Supplementary Fig. 9g). These findings suggest that the immune cell subpopulations that we had identified as associated with outcomes of cancer immunotherapy for melanoma also exist in BCC, and that the characteristic gene expression signatures may be altered similarly in melanoma and in BCC in the context of response to immunotherapy.

**The development of an ICT outcome signature**. Because the TREM2[hi] Mφ, Tgd_c21 and B_c22 populations exhibited the greatest quantitative differences between ICT non-responders and responders, we hypothesized that the expression of the feature genes of these populations may predict ICT outcome. To explore this hypothesis, we developed an ICT responsiveness signature based on the scRNA-seq dataset and a bulk gene expression dataset - GSE78220[9] using the *cancerclass* R package[23]. This signature had significantly high prognostic values for ICT outcomes in the discovery dataset. Specifically, for the GSE78220 dataset ($N = 28$, NR vs R: 13 vs 15), the signature had an AUC (Area Under The Curve) of 0.98 (95% confidence interval [CI],

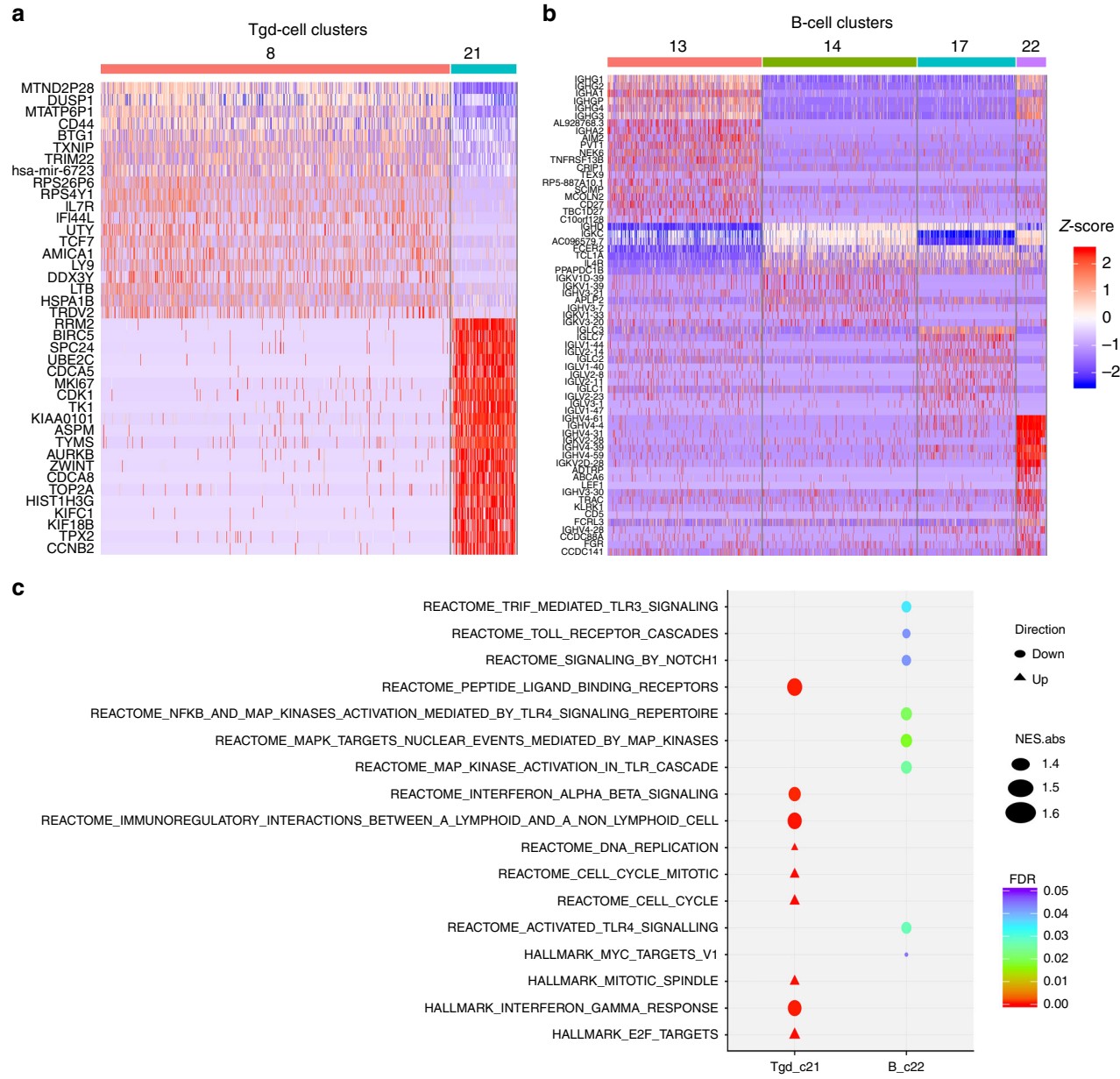

**Fig. 3 The analysis of the gammadelta T cells (Tgd) cells and B cells subsets in the melanoma samples.** The scRNA-seq dataset - GSE120575 was used in this analysis. **a** Heatmap of z-scored expression of the top up-regulated genes of the Tgd subpopulations – Tgd_c8 and Tgd_c21. **b** Heatmap of z-scored expression of the top up-regulated genes of the B-cells subpopulations – B_c13, B_c14, B_c17, and B_c22. **c** The significantly altered molecular pathways in the Tgd_c21 and B_c22 immune cell subpopulations whose percentages were associated with ICT outcomes.

0.96–1), sensitivity of 93% (95% CI, 72–100%), and specificity of 85% (95% CI, 59–97%; Fig. 4a). In the GSE78220 dataset, only one sample was early-on-treatment tumor and all the rest 27 melanoma samples are from pretreatment tumors. Because this ICT outcome signature was enriched for the characteristic genes of TREM2hi Mφ, Tgd_c21, B_c22 immune cell subpopulations (Supplementary Fig. 10), it was named as ImmuneCells.Sig. Detailed information of the genes of this signature can be found in Supplementary Data 9. Then the performance of ImmuneCells. Sig to predict ICT outcome was further validated using multiple independent bulk gene expression datasets of the pretreatment samples as follows.

To validate the above ICT response signature - ImmuneCells. Sig, we analyzed three independent gene expression datasets of melanoma patients to test the predictive performance of ImmuneCells.Sig[10–12]. For the first two datasets (GSE91061 and

PRJEB23709)[10,11], the pretreatment melanoma samples were selected for validation. Neither of these datasets were used to develop the ImmuneCells.Sig. For the GSE91061 dataset (N = 51, NR vs R: 25 vs 26), ImmuneCells.Sig performed well in differentiating NR from R tumors with an AUC of 0.96 (95% CI, 0.94–0.99), sensitivity of 88% (95% CI, 72–97%), and specificity of 92% (95% CI, 78–99%; Fig. 4b). For the PRJEB23709 dataset (N = 73, NR vs R: 27 vs 46), ImmuneCells.Sig also accurately predicted ICT outcomes: AUC of 0.86 (95% confidence interval [CI], 0.82–0.91), sensitivity of 78% (95% CI, 61–90%), and specificity of 78% (95% CI, 66–88%; Fig. 4c). The binomial confidence intervals for sensitivity and specificity were calculated by the Wilson procedure implemented in the *cancerclass* R package[23].

For further validation, we downloaded and analyzed the third dataset that includes the gene expression profile of a big cohort of

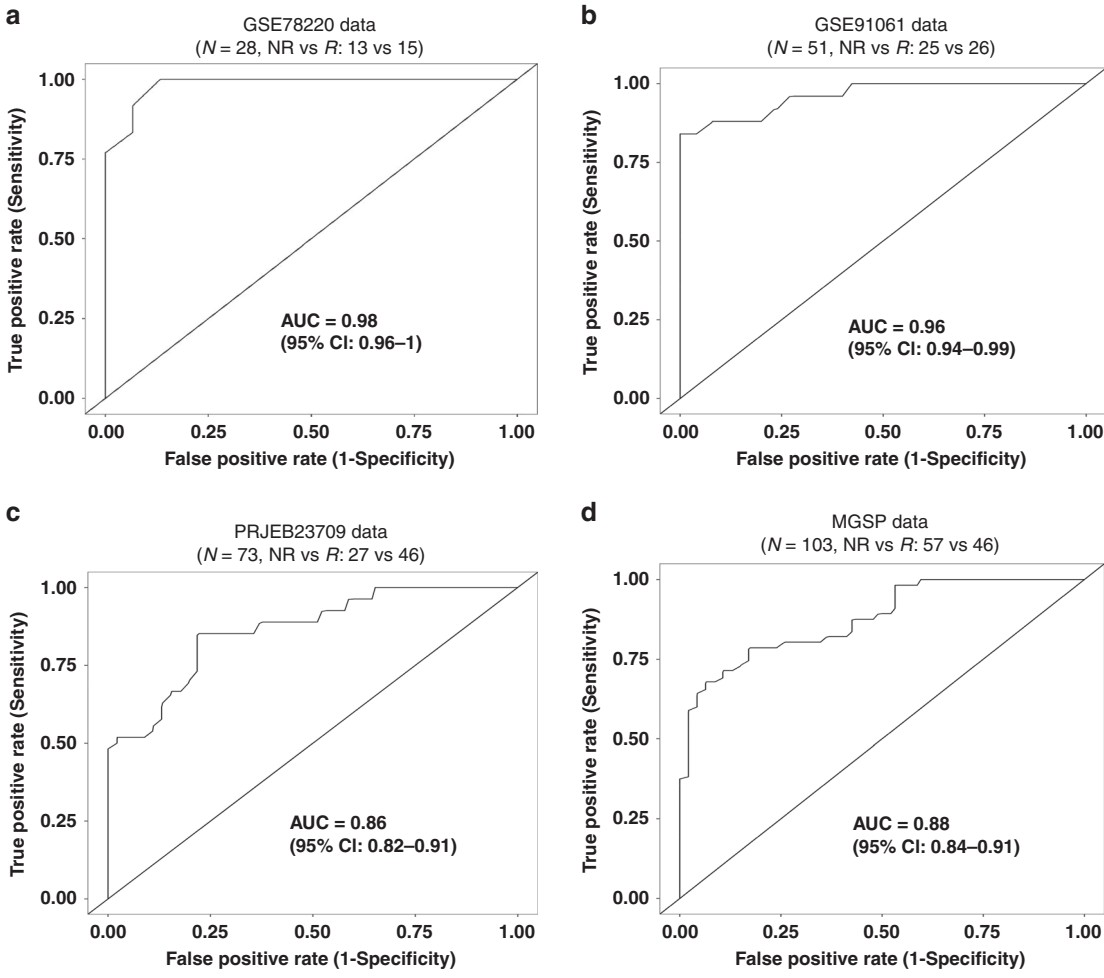

**Fig. 4 The ImmuneCells.Sig signature may predict ICT outcome in melanoma patients. a** ImmuneCells.Sig had significantly high prognostic values for ICT outcomes in the initial discovery dataset - GSE78220. **b** ImmuneCells.Sig accurately predicted the ICT outcome in the first validation dataset of GSE91061. **c** ImmuneCells.Sig accurately predicted the ICT outcome in the second validation dataset of PRJEB23709. **d** ImmuneCells.Sig accurately predicted the ICT outcome in the third validation dataset of MGSP.

melanoma patients who were treated by the anti-PD-1 immunotherapy, from which a large number of pretreatment melanoma samples from 103 patients with distinct response to ICT (46 responders vs 57 non-responders) had been subjected to RNA-seq[12]. Applied to this large dataset that was named as MGSP (melanoma genome sequencing project), the predictive value of ImmuneCells.Sig was still high. Specifically, it differentiated progressors from responders with an AUC of 0.88 (95% CI, 0.84–0.91), sensitivity of 79% (95% CI, 68–87%), and specificity of 79% (95% CI, 67–88%; Fig. 4d).

Among the four bulk RNA-seq datasets, only the PRJEB23709 dataset had pre-ICT biopsies for melanoma patients treated with either anti-PD-1 (41 patients: 19 non-responders vs 22 responders) or the combination of anti-PD-1 and anti-CTLA-4 drugs (32 patients: 8 non-responders vs 24 responders). We split the PRJEB23709 dataset into PRJEB23709_Pre_anti-PD-1 and PRJEB23709_Pre_Combo according to the treatment scheme (anti-PD-1 or combination of anti PD-1 and anti-CTLA-4). In each dataset, we tested the performance of ImmuneCells.Sig. It was found that ImmuneCells.Sig can accurately distinguish responders from non-responders in both Pre_anti-PD-1 and Pre_Combo subgroups. For PRJEB23709_Pre_anti-PD-1 subset, the performance of ImmuneCells.Sig is as follows: AUC = 0.88 (95% CI, 0.83–0.94), sensitivity = 86% (95% CI, 68–96%), and specificity = 79% (95% CI, 58–92%; Supplementary Fig. 11a). For

PRJEB23709_Pre_Combo subset, the performance of ImmuneCells.Sig is as follows: AUC = 0.93 (95% CI, 0.86–0.99), sensitivity = 88% (95% CI, 71–97%), and specificity = 88% (95% CI, 53–99%; Supplementary Fig. 11b).

Using the R package cancerclass, we can calculate the z-score in each pre-therapy biopsy based on the expression values of the ImmuneCells.Sig genes to predict who are more likely to respond to anti-PD-1 or anti-PD-1 plus anti-CTLA-4 combo therapy. For example, in the model built from Pre-anti-PD-1 dataset of PRJEB23709_Pre_anti-PD-1, the threshold z-score of 0.19 yielded sensitivity of 91% for responders. In the model built from Pre-Combo dataset of PRJEB23709_Pre_Combo, the threshold z-score of 0.1 yielded sensitivity of 91% for responders. Therefore, if we test a pre-therapy melanoma sample, the corresponding patient may not respond to either anti-PD-1 treatment or anti-PD-1 plus anti-CTLA-4 combo treatment if the z-score is <0.1, but may respond to the more toxic combo treatment if z-score is within the range of [0.1, 0.19], and may respond to the less toxic anti-PD-1 treatment alone if the z-score is >0.19. Therefore, prediction of the outcomes of different therapy regimen is possible based on the application of ImmuneCells.Sig.

To further evaluate the predictive performance of the ImmuneCells.Sig signature, we compared the ImmuneCells.Sig with the other 12 ICT response signatures reported previously (Table 1)[9,24–32], including the previously recognized IMPRES

**Table 1 The list of biomarkers for response to immune checkpoint therapy that were compared in this study.**

| Signature ID | Description |
|---|---|
| ImmuneCells.Sig | The immune cell signature identified in this study |
| IFNG.Sig | Interferon gamma (IFNγ) response biomarkers of 6 genes including IFNG, STAT1, IDO1, CXCL10, CXCL9, and HLA-DRA[24] |
| CD8.Sig | Gene expression level of CD8A + CD8B + CD3D + CD3E + CD3G[25] |
| PD-L1.Sig | Gene expression level of PD-L1 + PD-L2 + PD-1[25] |
| CRMA.Sig | Anti-CTLA4 resistance MAGE genes, including MAGEA2, MAGEA2B, MAGEA3, MAGEA6, and MAGEA12[26] |
| IMPRES.Sig | Immuno-predictive score (IMPRES), a predictor of Immune checkpoint blockade (ICB) response in melanoma based on 28 immune checkpoint genes[27] |
| IRG.Sig | A prognostic signature based on 11 immune-related genes (IRGs) for predicting CC (cervical cancer) patients' response to immune checkpoint inhibitors (ICIs)[28] |
| LRRC15.CAF.Sig | A signature of 14 marker genes of a specific type of carcinoma-associated fibroblasts (CAF) – "LRRC15+ CAFs" that correlated with poor response to anti-PD-L1 therapy[29] |
| T.cell.inflamed.Sig | An 18 gene "T-cell–inflamed gene expression signature" that can predict clinical benefit of anti-PD-1 in various cancer types (melanoma, head and neck squamous cell carcinomas, digestive cancers, ovarian and triple negative breast cancers)[24,30] |
| IPRES.Sig | IPRES (innate anti-PD-1 resistance) that included 16 genes involved in cell adhesion, extracellular matrix remodeling, angiogenesis, wound healing, and mesenchymal transition that predicted response to anti-PD-1 antibody therapy in melanoma[9,31] |
| Inflammatory.Sig | A gene expression signature of 27 inflammation related genes that predicted response to immune checkpoint blockade in lung cancer[31] |
| EMT.Sig | A gene expression signature of 12 epithelial-to-mesenchymal transition (EMT) related genes that predicted response to immune checkpoint blockade in lung cancer[31] |
| Blood.Sig | A blood sample based 15-gene expression signature that can predict response to anti-CTLA4 immunotherapy[32] |

signature[27]; they were all compared across the above four transcriptome-wide gene expression datasets of melanoma patients (i.e., the GSE78220, GSE91061, PRJEB23709, and MGSP datasets). The results show that the ImmuneCells.Sig was consistently the best signature for predicting response to immunotherapy across all four datasets (Fig. 5 and Supplementary Fig. 12). As a reference, the well-established IMPRES signature was ranked third in prediction accuracy in the GSE78220 dataset (Fig. 5a and Supplementary Fig. 12a), fifth in the GSE91061 dataset (Fig. 5b and Supplementary Fig. 12b), and second in both the PRJEB23709 and the MGSP datasets (Fig. 5c, d and Supplementary Fig. 12c, d). The fact that the ImmuneCells. Sig signature is the best predictor for the outcome of immune checkpoint therapy across the four independent melanoma datasets suggests that the ImmuneCells.Sig is an effective biomarker that can accurately predict ICT clinical outcome based on the pretreatment tumor samples from melanoma patients.

## Discussion
A large-scale single-cell RNA-seq study of tumor samples of melanoma patients treated by ICT[6] was re-analyzed to dissect individual cell populations that may correlate with response. Three immune cell clusters had drastically different percentages in ICT responders vs non-responders. The TREM2[hi] macrophages and Tgd_c21 T cells were markedly higher in the non-responders and could contribute to ICT resistance; in contrast, the B_c22 B cells were higher in the responders and could contribute to ICT anti-tumor response. TREM2[hi] macrophages, the most enriched immune cell subcluster in the non-responders, displayed a distinct gene expression pattern, with overexpression of key genes of the complement system. Expression of complement effectors and receptors has been associated with cancer progression and poor prognosis[33,34]. Among all the complement elements that may have the pro-cancer activities, C1q chains, C3-derived fragments, and C5a are likely the most important modulators of tumor progression[35,36]. In a clear-cell renal cell carcinoma (ccRCC) model, mice deficient in C1q, C4, or C3 displayed decreased tumor growth, whereas tumors infiltrated with high densities of C1q-producing macrophages exhibited an immunosuppressed microenvironment[37]. The classical complement pathway is a key inflammatory mechanism that is activated by cooperation between tumor cells and tumor-associated macrophages, favoring cancer progression[37]. Our findings extend this premise; TREM2[hi] macrophages, which overexpress major elements of the complement system and activation of the complement cascade, are enriched in ICT non-responders and could be the major macrophage subset that contributes to ICT resistance.

Although the role of complement system is not completely understood, other studies described different mechanisms by which complement activation in the tumor microenvironment can enhance tumor growth, such as altering the immune profile of tumor-infiltrating leukocytes, increasing cancer cell proliferation, and suppressing CD8 + TIL function[38]. More recently, complement effectors such as C1q, C3a, C5a, and others have been associated with inhibition of anti-tumor T-cell responses through the recruitment and/or activation of immunosuppressive cell subpopulations such as MDSCs (myeloid-derived suppressor cells), Tregs, or M2 tumor-associated macrophages (TAMs)[39]. The rationale of inhibiting the complement system for therapeutic combinations to enhance the anti-tumor efficacy of anti-PD-1/PD-L1 checkpoint inhibitors has been proposed based on the supporting evidence that complement blocks many of the effector routes associated with the cancer-immunity cycle[39]. Our study results were in line with these findings and suggest that the TREM2[hi] macrophage population which has an activated complement system could be another source or consequence of complement activation contributing to the blockade of cancer-immunity cycle.

Many M2 polarization genes, some of which are known to be tumor-promoting, were also overexpressed in TREM2[hi] macrophages. For example, CD276 (B7-H3) plays a role in down-regulating T-cells involved in tumor immunity[40,41]. High CD276 expression is associated with increased tumor size, lymphovascular invasion, poorly differentiated tumors, and shorter overall patient survival[42,43]. CD276 expression is also associated with tumor-infiltrating FOXP3 + regulatory T cells which inhibit effector T cells[44,45] and is important for immune evasion and tumorigenesis in prostate cancer[46]. CD276 also inhibits NK cell lysis of tumor cells[47]. The overexpression of CD276 in TREM2[hi] macrophages likely has implications for promoting ICT resistance. PD-L2, an important immune co-inhibitory molecule[48], was also overexpressed in the TREM2[hi] macrophages. Increased expression of PD-L2 in tumor-associated macrophages

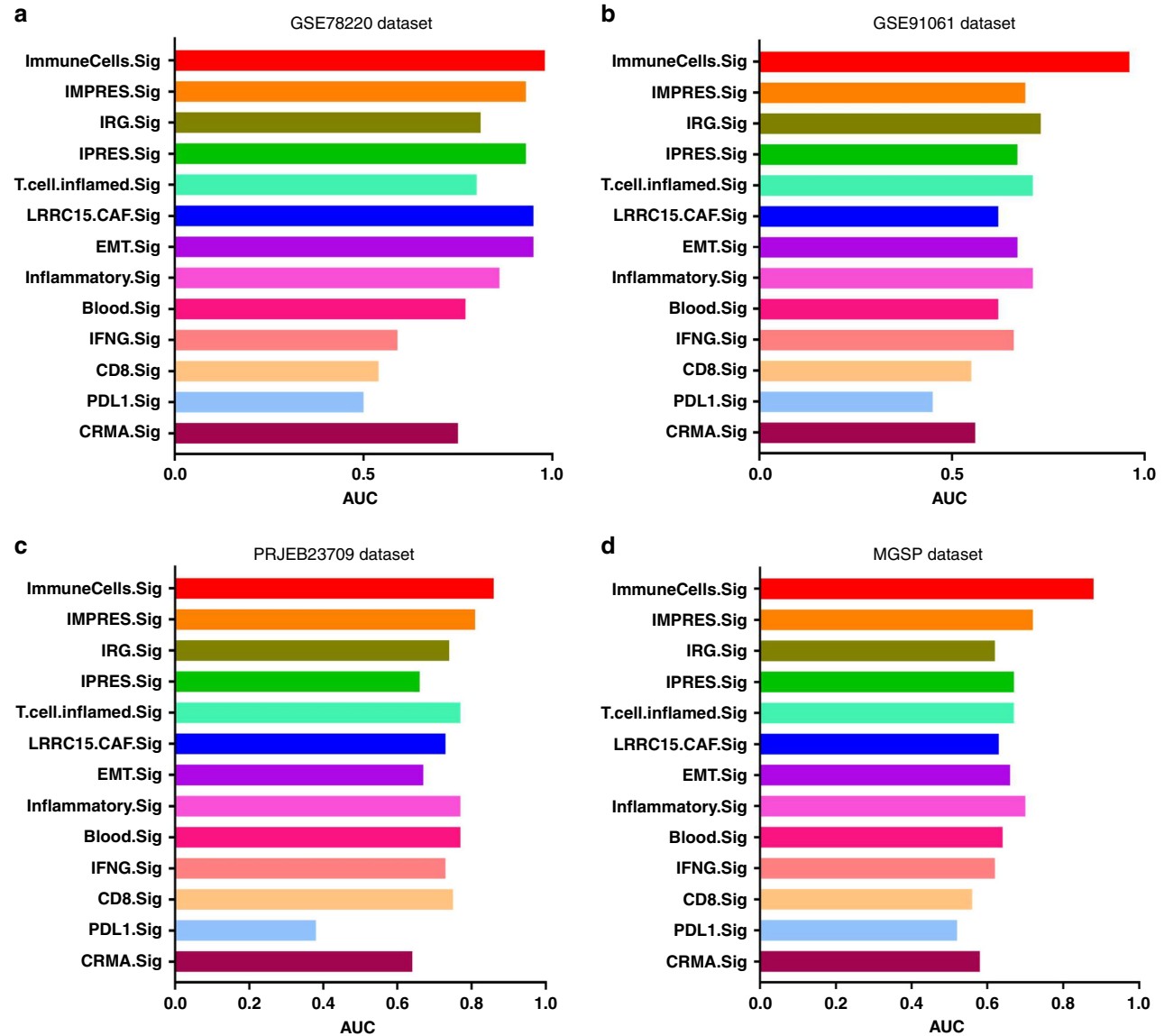

**Fig. 5 Comparison of the performance of ImmuneCells.Sig with other ICT response signatures.** The multiple barplots for the AUC values of the 13 ICT response signatures are shown in **a** for the GSE78220 dataset. **b** for the GSE91061 dataset. **c** for the PRJEB23709 dataset. **d** for the MGSP dataset.

contributes to suppressing anti-tumor immunity in mice treated with anti-PD-L1 monoclonal antibody[49]. Thus, the high PD-L2 expression in TREM2[hi] macrophages could facilitate ICT resistance and tumor progression. Some single-cell studies reported that M1 and M2 signatures are positively correlated in myeloid populations[50,51]. We checked the expression of M1 markers from these studies in the TREM2[hi] macrophages (Supplementary Fig. 13). It was found that the expression of M1 signature genes was neither strong nor consistent. The gene - *iNOS* (*NOS2*), the most characteristic and canonical M1 macrophage marker[20,50–52] was not expressed in the TREM2[hi] cell population (Supplementary Fig. 13). These results suggest that TREM2[hi] macrophages are functionally more proximal to M2 polarization macrophages. TREM2[hi] macrophages had been reported in a breast cancer single-cell study to be a branch of recruited or resident M2 type macrophages expressing several genes in common with our study such as *SSP1*, *C1Q*, *CCL18*, and *MACRO*[50]. However, TREM2[hi] macrophages had not been linked to cancer immunotherapy response before. So that aspect of our data is valuable to clinical practice in cancer immunotherapy.

A γδ T cells subset, Tgd_c21, was present at much higher levels in the non-responders. Despite their role in anti-tumor cytotoxicity, γδ T cells could also promote cancer progression by inhibiting anti-tumor responses and enhancing cancer angiogenesis. Consequently, γδ T cells have a dual effect and are considered as being both friends and foes of cancer[53]. The enrichment of the Tgd_c21 cells in the ICT non-responders suggests an association with ICT resistance. The top Tgd_c21 marker genes are oncogenic by nature including *RRM2*[54], *BIRC5* (*Survivin*)[55], *SPC24*[56,57], *UBE2C*[58,59], and *CDCA5*[60]. Pathway analysis revealed a significant reduction in ligand-receptor binding capacity, IFNα and IFNβ signaling, IFN-γ response, and immunoregulatory interactions of Tgd_c21 cells, suggesting that Tgd_c21 cells may be a type of 'exhausted' γδ T cell with impaired anti-tumor immune functions. A previous study showed that the positive outcome of PD-1 blockade on treating leukemia may be because that it induces significant upregulation of the potent pro-inflammatory and anti-tumor cytokine IFN-γ in certain types of γδ T cells[61]. Complementing their study, we showed that the failure of immunotherapy in treating melanoma may be

associated with some types of γδ T cells (e.g., Tgd_c21). The pathway analysis showed that this subset of γδ T cells - Tgd_c21 had decreased activity of the anti-tumor IFN-γ pathway in the non-responders than the responders subjected to the immunotherapy (Fig. 3c). Therefore, a key element may be the IFN-γ pathway activity, whose reduction in some γδ T-cell subsets such as Tgd_c21 in ICT non-responders may contribute to ICT resistance.

All B-cell clusters were depressed in the ICT non-responders. Apart from their role in antibody production, B cells also are an important source of cytokines and chemokines that contribute to anti-tumor immune responses[62]. Therefore, the decreased B-cell percentages in non-responders could contribute to ICT resistance and/or progression of ICT-resistant tumors. We compared the present B-cell subpopulation signature (B_c22, derived from Supplementary Data 8 based on cutoff *P* value 0.05) with the other B-cell signature recently published in the context of ICT by Helmink et al.[63] and found several genes shared by both signatures including *TCL1A*, *ITIH5*, *LAX1*, *KCNA3*, *CD79A*, *AREG*, *GBP1*, *ATP8A*, and *IGLL5*. Both our signature and their signature characterized the B-cell populations that were significantly enriched in the ICT responders versus non-responders. However, the B cells associated with these two signatures were different. This is because our B_c22 (single cell cluster 22) signature was developed based on the scRNA-seq data of melanoma samples and its corresponding B cells were a subset of B cells that were highly enriched in the ICT responders than the non-responders. We also identified three other B-cell subpopulations corresponding to clusters 13, 14, and 17 (Fig. 1 and Supplementary Table 1). In contrast, the B-cell signature used by Helmink et al.[63] was derived from bulk RNA-seq data of renal cell carcinoma (RCC); thus, their signature may represent a mix of B-cell subpopulations enriched in RCC patients that responded to ICT. Therefore, it is logical for the two signatures to share some, but not all genes.

For comparison with ImmuneCells.Sig, we used the gene signature representing the three component cell clusters (TREM2[hi] macrophages, Tgd_c21 γδ T cells, and B_c22 B cells) identified from the scRNA-seq data (Figs. 2 and 3 and Supplementary Fig. 7). This 150-gene signature is composed of three sets of top 50 genes most significantly over-expressed in one of the three cell clusters (the top ranked 50 genes from the Supplementary Data 1, 2, and 3). This gene signature was called scR.Immune and used for ICT outcome prediction. The scR.Immune signature had a somewhat lower predictive capability compared with the ImmuneCells.Sig signature derived from both scRNA-seq and bulk gene expression datasets. As seen in Supplementary Fig. 14, the AUC values from scR.Immune were 0.92, 0.90, 0.84, and 0.78 for the datasets of GSE78220, GSE91061, PRJEB23709, and MGSP, respectively, which are lower than the AUC values given by the ImmuneCells.Sig signature (0.98, 0.96, 0.86, and 0.88 for the four datasets, respectively). The difference in predictability between these two sets of signatures is likely due to the complex cellular composition of tumors. Because the four datasets used for AUC calculations are all bulk gene expression data, the corresponding expression levels of genes represented a mix of expression from all kinds of cells embedded in the tumor samples. Therefore, using scRNA-seq data derived signature alone such as the scR. Immune signature may not predict ICT outcome better than using the ImmuneCells.Sig signature derived from both scRNA-seq and bulk gene expression datasets. However, the ImmuneCells.Sig signature is enriched for the signature genes from the TREM2[hi] macrophages, Tgd_c21 γδ T cells and B_c22 B cells, suggesting the involvement of these immune cell subpopulations in determining the ICT responsiveness. This signature may also be useful to predict progressive versus responsive melanoma tumors extracted from the same patients treated

with ICT[64]. A limitation of this study is that deciphering the biological meanings of the above relevant cell types that impact the efficacy of ICT treatment remains unsolved. Well-designed experimental strategies should be used to explore the hidden mechanisms to strengthen the biological findings of this study.

The decreased percentage of B cells and increased percentage of macrophages/monocytes in ICT non-responding patients had been reported previously[6]. However, the important subsets of these immune cell populations were not revealed as in this study. Moreover, we identified an ICT outcome gene expression signature, ImmuneCells.Sig, that is enriched for the characteristic genes of TREM2[hi] macrophages, Tgd_c21, and B_c22 subpopulations. The ImmuneCells.Sig signature outperformed the other outstanding signatures in predicting the outcome of immune checkpoint therapies across all four independent datasets[9–12]. Our characterization of these immune cell populations provides the opportunities to improve the efficacy of cancer immunotherapy and to better understand the mechanisms of ICT resistance.

## Methods

**Study design.** Single-cell RNA-sequencing data (accession number GEO: GSE120575) of melanoma samples from the initial publication[6] were down-loaded and re-analyzed for this manuscript. For the validation purposes, two other scRNA-seq datasets[7,8] of melanoma and BCC were also downloaded, which are accessible through GEO accession number: GSE115978 and GSE123813. For the development of the ICT outcome signature, we analyzed the transcriptome-level gene expression data set (GSE78220) of an immune checkpoint therapy (ICT) study[9]. For the validation of the identified ICT outcome signature - ImmuneCells. Sig, we analyzed three large public gene expression datasets of immunotherapy[10–12] (respectively accession number: GSE91061, ENA project PRJEB23709, dbGaP phs000452.v3.p1). The first dataset[10] (GSE91061) consisted of pretreatment melanoma samples from 51 patients (25 non-responders and 26 responders). For the second dataset[11] (PRJEB23709), the scRNA-seq data of the 73 pretreatment tumors were analyzed. Among these 73 samples, 41 are from the melanoma patients subjected to anti-PD-1 therapy and consist of 19 non-responders and 22 responders; 32 are from the melanoma patients subjected to combined anti-PD-1 and anti-CTLA-4 therapy and consist of 8 non-responders and 24 responders. The third dataset (phs000452.v3.p1) is from a large melanoma genome sequencing project (MGSP)[12] from which the whole-transcriptome sequencing (RNA-seq) data from 103 pretreatment tumor tissue samples from 103 patients with distinct ICT outcomes (47 responders and 56 non-responders) were available and used for validation in this study.

**Single-cell RNA sequencing data analysis.** The data from a previous scRNA-seq study of melanoma checkpoint immunotherapy[6] were analyzed. Specifically, we utilized the Seurat v3.0 R package[13,14] to perform the fine clustering of the 16,291 single cells. The gene expression data from single cells of both conditions, i.e., regression/responder (R group: n.patients = 17; n.cells = 5564) and progression/non-responders (NR group: n.patients = 31; n.cells = 10,727), were aligned and projected in a 2-dimensional space through uniform manifold approximation and projection (UMAP)[16] to allow identification of ICT-outcome-associated immune cell populations. Highly variable genes – genes with relatively high average expression and variability – were detected with Seurat[13,14]. These genes were used for downstream clustering analysis. Principal component analysis (PCA) was used for dimensionality reduction and the number of significant principal components was calculated using built in the JackStraw function. t-distributed stochastic neighbor embedding (t-SNE) and UMAP were used for data visualization in two dimensions.

The built-in FindMarkers function in the Seurat package was used to identify differentially expressed genes. From results of the Seurat package, genes with adjusted *P* values < 0.05 were considered as differentially expressed genes. Adjusted *P* values were calculated based on Bonferroni correction using all features in the dataset following Seurat manual [https://satijalab.org/seurat/v3.0/de_vignette. html]. Genes retrieved from Seurat analysis were displayed in heatmap using scaled gene expression calculated with the Seurat-package built-in function. Fold change plots were created in R with ggplot2 package. For the two scRNA-seq data[7,8] of melanoma and BCC that were used for validation, i.e., GSE115978 and GSE123813 datasets, the pre-processed gene expression data were downloaded, processed, and analyzed in the same way as done for the discovery scRNA-seq dataset - GSE120575.

**RNA-seq data and ICT responsiveness signature analysis.** For the bulk RNA-seq datasets[9–11], we processed them in the following steps. The downloaded FASTQ files containing the RNA-seq reads were aligned to the hg19 human genome using Bowtie-TopHat (version 2.0.4)[65,66]. Gene-level read counts were

obtained using the htseq-count Python script from HTSeq v0.11.1 [https://htseq.readthedocs.io/en/release_0.11.1/] in the union mode. We further utilized the iDEP v0.92[67] [http://bioinformatics.sdstate.edu/idep/] to transform the read counts data using the regularized log (rlog) transformation method originally implemented in the DESeq2 v1.28.1 package[68] [https://bioconductor.org/packages/release/bioc/html/DESeq2.html], as it effectively reduces mean-dependent variance. The transformed data are used for the downstream analysis and available as detailed in the Data availability statement.

Because three single-cell clusters – TREM2hi macrophages, Tgd_c21, and B_c22 exhibited large quantitative changes between the ICT responders and non-responders, we hypothesized that the tumor expression of the feature genes of these specific immune cell populations may be useful to predict the ICT outcome. In order to test this hypothesis, we developed an ICT responsiveness signature based on the scRNA-seq dataset and a bulk gene expression dataset – GSE78220[9] using the *cancerclass* R package[23]. To validate this ICT response signature – ImmuneCells.Sig, we analyzed three independent gene expression datasets of melanoma patients[10–12] (GSE91061, PRJEB23709, and MGSP datasets) and corroborated the high prediction values of ImmuneCells.Sig. We also compared the ImmuneCells.Sig with the other 12 ICT response signatures reported previously (Table 1)[9,24–32] across the above four gene expression datasets of melanoma patients. The corresponding R codes are available as detailed in the Code availability statement.

**Pathway analyses**. Pathway analyses were conducted using several excellent software tools, including IPA software (IPA release June 2020, QIAGEN Inc., [https://www.qiagenbioinformatics.com/products/ingenuitypathway-analysis]), Gene Set Variation Analysis[69] (GSVA v1.36.2, [https://bioconductor.org/packages/release/bioc/html/GSVA.html]), and Gene Set Enrichment Analysis[22] (GSEA v4.0.0, [https://www.gsea-msigdb.org/gsea/index.jsp]). GSEA analysis was performed for pre-ranked differentially expressed genes using the option - Gsea-Preranked. One thousand permutations were used to calculate significance. A gene set was considered to be significantly enriched in one of the two groups when the raw $P$ value < 0.05 and the FDR (false discovery rate) was <0.25 for the corresponding gene set. In addition, we utilized an R-package called Fast Gene Set Enrichment Analysis (fgsea v1.15.1, [https://github.com/ctlab/fgsea]). The package implements a special algorithm to calculate the empirical enrichment score null distributions simultaneously for all the gene set sizes, which allows up to several hundred times faster execution time compared to original Broad implementation of GSEA. Reactome pathways analyses were performed using Protein ANalysis THrough Evolutionary Relationships (PANTHER v15.0, [http://pantherdb.org/]). The associated settings are - Analyze type: PANTHER Overrepresentation Test, release 20190711; Annotation Version and Release Date: Gene Ontology database Released 2019-07-03 [http://geneontology.org/]) with lists of significantly enriched genes in the corresponding clusters as detected by Seurat.

**Statistical analysis**. The performance of the ImmuneCells.Sig as a classifier for ICT outcome was evaluated with the use of receiver-operating-characteristic curves (ROC), calculation of AUC (Area under the ROC Curve), and estimates of sensitivity and specificity implemented in the *cancerclass* v1.32.0 R package[23]. This classification protocol starts with a feature selection step and continues with nearest-centroid classification. The binomial confidence intervals for sensitivity and specificity were calculated by the Wilson procedure implemented in the *cancerclass* R package[23]. Fisher's exact test was used for categorical variables. All confidence intervals are reported as two-sided binomial 95% confidence intervals. Statistical analysis was performed with R software, version 3.5.3 (R Project for Statistical Computing).

**Reporting summary**. Further information on research design is available in the Nature Research Reporting Summary linked to this article.

## Data availability

All of the datasets used in this study have been obtained from publicly available sources. Gene Ontology database Released 2019-07-03 [http://geneontology.org/] was used in pathway analyses. The scRNA-seq datasets[6–8] were retrieved from Gene Expression Omnibus (GEO) under accession number GSE120575, GSE115978, and GSE123813. Transcriptome-level gene expression data sets of immune checkpoint therapy studies[9–11] were retrieved under accession number GSE78220, GSE91061, and from ENA project PRJEB23709. A dataset named as MGSP (melanoma genome sequencing project), containing data from a large melanoma genome sequencing project[12] is available in dbGaP under accession number phs000452.v3.p1. All other relevant data are available in the article, supplementary information, or from the corresponding author upon reasonable request.

## Code availability

The computer R codes for the scRNA-seq and gene signature analysis are available at https://github.com/donghaixiong/Immune_cells_analysis.

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

## Acknowledgements

This work was supported by the US National Institutes of Health (NIH) (R01CA223804). We thank Drs. Bryon D. Johnson, Matthew J. Riese, Weiguo Cui and Charles R. Myers for their careful and critical reading of this manuscript.

## Author contributions

M.Y. and Y.W. conceived of the project and revised the manuscript. D.X. performed the data analyses and writing of the paper.

## Competing interests

M.Y. is a co-founder of OncoImmune, Inc. All other authors declare no competing interests.
