## [Peer Review File · Nature Communications]

Reviewers' comments:

Reviewer #1 (Remarks to the Author):

This manuscript identifies as its key issue to use single-cell RNA sequencing to identify factors underlying ICT resistance. It claims to have a signature that provides "one of the most accurate predictors to date of ICT response". However, upon my reading the validation claimed was not so clear. If the punch line was to be Figure 5, why are there so many AUCs given in Figure 6. If a 40-gene solution was presented as near ideal, what was all the remaining work seeking to show. This manuscript thus did not provide a clear presentation to this reviewer to really make sense of the findings claimed.

Specific issues:

1. For the binomial confidence intervals presented, such as for sensitivity and specificity on page 8, please provide the statistical method used.
2. (p 12) Change "Principle Components" to "Principal Components" in two locations.
3. (p. 12) Identify the statistical method used for p-value adjustment.
4. (Results and Table S1) Three clusters are identified as having Fold rates of 6 or higher in absolute value, based on a cutoff from a cited paper. It is interesting that only four clusters had percentages of < 0.50 for one of their R of NR groups, and three were the significant clusters. Please provide some further defense that this approach to identifying clusters of interest is justified.
5. (Figure 4). Were the p-values shown on the y-axis adjusted or not? What is being plotted on the x-axis?
6. (Figure S1) Can confidence interval be given to the bars in the figure. If so, please do so. Please include the denominator that applies to the percentages.
7. Please spell out what IPA analysis is when first presented.
8. Throughout the paper (e.g. p 5, bottom, and Figure S2), please indicate whether a p-value is a raw p-value or an adjusted p-value. If it is a raw p-value, and an adjustment is needed to properly interpret the p-value, please provide it as well.
9. Something seems deeply flawed statistically in the very long eSpreadsheets which generate adjusted p-values that are so strikingly low (e.g. 3 and one-half pages of genes whose adjusted p-values are less than E-50.) It is unclear that such long lists produce any clear information. If they do, perhaps some greater explanation of them is needed.

Reviewer #2 (Remarks to the Author):

The study by Xiong et al. focused on one public single-cell RNA-seq dataset of ICT-treated melanoma patients, and reanalyzed the immune cell populations in two groups of patients with different ICT outcomes. The authors compared the immune cell clusters between responders and non-responders, and discovered specific subsets of macrophages, $\gamma\delta$ T cells and B cells that were potentially correlated with ICT resistance in melanoma. Although the authors claimed to have revealed an immune signature that could predict ICT outcomes, this study in the end were highly descriptive, and the major discoveries were all based on very limited analyses. In other words, the reliability of the conclusions of this study can be improved, and further analyses and experiments are needed to support the biological findings.

Major concerns,

1. It is difficult to know how robust the clustering result is, especially for the heterogeneity of macrophages, since there is little to no direct testing of the predicted key functionally, mechanistically or clinically relevant features of different subpopulations.
2. The authors compared the percentages of each cell clusters in the integrative clustering result to differentiate cell frequencies between responders and non-responders, and drew the major

conclusions of this study. However, this dataset involves 48 samples and each group contains enough number of samples to perform statistical analysis, and the authors should perform statistical tests to increase the credibility of such analyses.

3. Similarly, it is not appropriate to only utilize the overall fold differences of percentages of cell clusters in two patient groups to conclude the enrichment or deficiency of specific cell clusters in the resistant group (Fig 2B), since fold differences of cell frequencies are highly affected by the size of the cell frequencies of pre-treatment. Thus, statistical analyses should be performed at the patient level.

4. The authors discovered that TREM2hi macrophages were enriched in ICT resistant group and assumed that this subset were functionally proximal to M2 polarization macrophages, which could block the anti-tumor activities of ICT. However, the single-cell studies by Azizi et al. (Cell, 174, 1293-1308 e36, 2018) and Wagner et al. (Cell 177, 1330-1345 e18, 2019) reported that M1 and M2 signatures are positively correlated in myeloid populations. The authors should check the expression of M1 markers in TREM2hi population. In addition, TREM2hi macrophages highly expressed complement system genes and enriched complement activation pathway, the function of which in anti-tumor immunity seems controversial (Ann Transl Med, 4(14): 265. 2016; J Clin Invest. 2017 Mar 1; 127(3): 780-789.2017). Thus, the authors should experimentally confirm the function of TREM2hi macrophages.

5. The biological meaning of the enrichment of one specific subset of $\gamma\delta$ T cells in non-responder patients should be explored. Although the authors discovered that this subset of cells had reduced ligand-receptor binding capacity, IFN α and IFN β signaling, IFN- γ response, and immunoregulatory interactions, it is not enough to deduce their roles in mediating ICT resistance.

6. The authors identified a gene signature that could predict ICT outcomes from bulk data, and claimed that this signature was enriched for the characteristic genes of the three significant single-cell clusters - TREM2hi macrophages, Tgd_c21 $\gamma\delta$ T cells and B_c22 B cells. What is the predictive capability of the gene signature of these three cell clusters directly identified from single-cell data? And what's the difference between this two sets of signatures derived from different data resources?

Other points,

1. The result showed in barplot of Fig 2A could be improved with boxplot showing the variations of patient level.

2. The authors should confirm the specificity of the 40-gene set identified to characterize TREM2hi macrophages to ensure the reliability of the confirmation analysis in bulk data.

3. It is better to move Fig 3A and Fig 4 to supplementary figures.

4. IDO1 was reported to be one of the major markers of IDO1hi populations of macrophages, however, the violin plot in Fig 3C showed that this gene was hardly expressed in IDO1hi subset. Maybe the authors could rename this subpopulation with other specific genes.

5. It is better to show the enrichment of predictive signature in markers of three cell clusters with other forms of figure resembling GSEA analysis, rather than to show the expressions of selected markers in each cluster (Fig 5B).

Reviewer #3 (Remarks to the Author):

In summary, this work reports the identification of a new single-cell-based RNA signature that could be predictive of clinical outcome for patients treated with immune checkpoint therapies (ICT) for melanoma. The authors have worked on the dataset from a previously published work by Sade-Feldman et al (reference 17) where 48 biopsies from patients treated with ICT were analyzed by scRNAseq. Rather than focusing on CD8 lymphocyte as in the previous study, they study other immune cell compartment, such as myeloid population, rare $\gamma\delta$ T cells and B cells.

23 clusters of immune cells were individualized based on the sc-RNAseq and 3 of these clusters showed more than 6 fold differences between NR and R patients. Clusters 12 and 21 were more represented in NR patients and corresponded respectively to a subset of macrophages and to proliferating $\gamma\delta$ T cells, whereas cluster 22, a B cell cluster was less represented in NR samples. This was done initially on all samples, then analyzed separately with regards to pre or post ICT as well as to the type of ICT regimens, where the same tendencies were observed (although on extremely small samples of patients).

More detailed characterization of cluster 12 showed that it was related to a TREM2^{high} population of macrophages expressing M2 polarization genes as well as several complement system genes. A TREM2 macrophage signature of 40 genes was designed and found to be associated with NR when interrogating publicly available patients' RNA signatures.

A neosignature named ImmuneCells.Sig, containing over 90 genes integrating scRNA variations and bulk RNA data from a previous analysis (GSE78220) was defined. Unsurprisingly it was well correlated to the outcome when tested on the patients from the GSE78220 since this data was used to create the signature. Finally, the authors find that this ImmuneCells.Sig compares favorably to other signatures that are associated with ICT response to date.

Comments & questions:

Patients' material:

1. Drawing general conclusions based on a mix of pre- and post treatment biopsies and mixing various treatment regimens does not make a lot of sense when one aims at identifying predictive biomarkers because the predictive signature can be used in the clinic only when analyzing pre-treatment biopsies (or early on-treatment biopsies). We must thus acknowledge the efforts done by the authors in performing differential analyses on pre-treatment samples, then on post-treatment samples or on samples from patients with different types of treatments. But then, we see that we only have a very small group of patients in the pre-ICT situation: 8 non-responders versus 4 responders before anti PD1. All the other biopsies that could be analyzed were taken post-ICT. Of course, the immune infiltrate of these post-ICT samples could have been modified by the treatment. This does not mean that it is not interesting to study but it addresses a question distinct from the identification of predictive baseline markers. How do the authors explain that data found on such a small number of pre-therapy samples could be useful to derive a pre-treatment signature?

2. Sup fig 1A is very misleading and cannot be used to make the point the authors want to make. Indeed, GSE79691 data is not about 10 patients but reports the results from 10 biopsies taken post-mortem from the same patient. This post-therapy data is thus not predictive of response but shows the heterogeneity of the immune response to ICT in various metastases from the same patient.

3. For the data from PRJEB23709: what are the 27NR and 46 R from the Gide et al manuscript : anti PD1 or combination? Pre and post therapy?

4. Could the authors show cohorts data that are from pre-ICT biopsies and using only anti PD1 or only the combination of anti PD1 and anti CTLA4? This would help to know if the new signature could have a real application in the clinic. Who to treat based on pre-therapy biopsy? Who to treat with the toxic combination regimen anti CTLA4 + anti PD1 versus the less toxic anti PD1?

Concerning the definition and interpretation of the scRNA signatures results:

1. With regards to the previous published study by Sade-Feldman et al: the authors should

mention that the increased B cells and macrophages in NR patients is not a new data as it was already shown in the initial paper. Figure 1 and 2 here do not bring two different message and could be presented in one figure.

2. How do the authors explain that the CD8+ T cells are found less represented in their analysis in responding patients whereas it was found increased in the initial paper? This is an obvious discrepancy that is not mentioned in the manuscript.

3. There is a concern about the definition of $\gamma\delta$ Tcells based on the combination CD3+CD4-CD8- only that could also correspond to NK cells that are in cluster 15, quite close to clusters 8 and 21 and can also be CD4-CD8-. Is the unique constant region of the TCR δ chain (TRDC) over-represented in the cluster 8 and 21? Does this population express the receptors TCRV δ 1 or TCRV δ 2 ? Altogether, the definition of what is here considered to be $\gamma\delta$ Tcells should be validated by the expression of additional $\gamma\delta$ Tcells signatures published in ImmGen and with published gene signatures of $\gamma\delta$ T cells (for example PNAS June 11, 2019 116 (24) 11906-11915).

4. How does the present B cell signature compare with data on B cells recently published by several teams in the context of ICT? This was not discussed in the text.

5. The authors find that the B cell cluster 22 has reduced oncogenic pathways expression. How could this explain that they could contribute to the activation of the oncogenic signaling in the tumor environment? This seems as gratuitous suggestion with no evident logical explanation, and no mechanism explored.

6. Additional datasets of scRNAseq are available, in particular, the one published in cell by Jerby-Arnon et al (<https://doi.org/10.1016/j.cell.2018.09.006>). Using this additional data should be done as it could strengthen the results.

7. The presence of certain immune cells, especially macrophages and gamma-delta-T cells could very well depend on the type of cancer or the type of organs involved. This origins of the metastases are thus important to mention and to analyze with regards to the type of infiltrate. It would also be interesting to look at various tumor types and not only melanoma patients to see if the results can be generalized to other types of cancers treated with ICT.

8. TREM2hi macrophage were already reported in breast cancer single cell studies (Azizi et al, Cell, 2018) to be a branch of recruited or resident M2 type macrophages expressing several genes in common with present study: SSP1, C1Q, CCL18, MACRO. This is thus not a new data.

In Conclusion: This work reports descriptive data derived from a secondary analysis of a scRNAseq dataset and leading to a new predictive signature for ICT efficacy that seems more robust than other RNA signatures in the same context. The definition of certain clusters is suboptimal and no mechanistic data are presented. Several hypotheses and statements are not substantiated by clear data. The signature seems interesting although its use remains in the research domain and not yet applicable in the clinic.

Reviewers' comments:

Reviewer #1 (Remarks to the Author):

This manuscript identifies as its key issue to use single-cell RNA sequencing to identify factors underlying ICT resistance. It claims to have a signature that provides “one of the most accurate predictors to date of ICT response”. However, upon my reading the validation claimed was not so clear. If the punch line was to be Figure 5, why are there so many AUCs given in Figure 6. If a 40-gene solution was presented as near ideal, what was all the remaining work seeking to show. This manuscript thus did not provide a clear presentation to this reviewer to really make sense of the findings claimed.

Reply: Sorry for this confusion, the 40-gene set was used as marker gene set for TREM2^{hi} macrophages. We discussed it because the activity of this geneset is higher than other macrophage subsets (Supplementary Figure 7D) and its activity score is also higher in the melanoma of non-responders than responders (Supplementary Figure 7B, 7C). However, we do not use it as predictive signature for ICT response. The ImmuneCells.Sig was the predictive signature built on the analysis of scRNAseq and bulk RNAseq data, which was enriched for the marker genes of the immune cell subsets we highlighted in this manuscript including the TREM2^{hi} macrophages, the Tgd_c21 $\gamma\delta$ T cells and the B_c22 B cells. Therefore, the AUCs were plotted for the ImmuneCells.Sig. We made revisions according to this and other comments so the presentation is clearer now.

Specific issues:

1. For the binomial confidence intervals presented, such as for sensitivity and specificity on page 8, please provide the statistical method used.

Reply: We used the following statistical method and have added this description in the Method section of the revised manuscript:

“The binomial confidence intervals for sensitivity and specificity were calculated by the Wilson procedure implemented in the *cancerclass* R package.¹”

2. (p 12) Change “Principle Components” to “Principal Components” in two locations.

Reply: This has been changed in the corresponding locations. Thank you.

3. (p. 12) Identify the statistical method used for p-value adjustment.

Reply: According to the Seurat software manual, the adjusted p-values were calculated based on Bonferroni correction using all features in the dataset (https://satijalab.org/seurat/v3.0/de_vignette.html). We have added this statement in the revised manuscript.

4. (Results and Table S1) Three clusters are identified as having Fold rates of 6 or higher in absolute value, based on a cutoff from a cited paper. It is interesting that only four clusters had percentages of < 0.50 for one of their R of NR groups, and three were the significant clusters.

Please provide some further defense that this approach to identifying clusters of interest is justified.

Reply: Our approach is exactly the same approach used in the previous scRNA-seq study² of the effects of the immunotherapy on changing the percentages of different immune cell subpopulations, which was published in *Cell* (*Cell*. 2018;175(4):1014-1030 e1019; PMC6501221). They compared the percentage of cells in individual clusters for different conditions of control, anti-PD-1, anti-CTLA-4, and anti-PD-1/anti-CTLA-4 (Figure 2D, 3F, 4D, 5D of the *Cell* paper).² In this way, they identified a number of significant immune cell subclusters that could be associated with the variation of the immunotherapy efficacy. In addition, we performed statistical tests (Wilcoxon tests) at the patient level for each of the 23 immune cell clusters based on two groups containing 17 samples for the responsive group and 31 samples for the non-responsive group. The results (Supplementary Figure 2) supported the results of comparing the percentages of each cell cluster in the integrative clustering to differentiate cell frequencies between responders and non-responders. We added the above justification in the manuscript.

5. (Figure 4). Were the p-values shown on the y-axis adjusted or not? What is being plotted on the x-axis?

Reply: (Figure 4 has become Supplementary Figure 6 in the revision to address other comment) The p-values shown on the y-axis had been adjusted by Bonferroni correction. We clarified this in the legend in the revision. The terms in the 'Reactome pathways' were plotted on the x-axis. This is the same approach as adopted in the previous single-cell RNA-seq study that performed gene ontology (GO) term analyses on different types of immune cell subpopulations³, which was published in *Circulation Research* (*Circ Res*. 2018;122(12):1661-1674. PMID: 29545365). Our Figure is similar to Figure 3 (A, B, C) and Figure 4A of that study.³

6. (Figure S1) Can confidence interval be given to the bars in the figure. If so, please do so. Please include the denominator that applies to the percentages.

Reply: The bars showed the real percentages of cells in each subcluster, not an estimate, so there is no confidence interval to be calculated. Our calculations were done in the same way as reported in the previous outstanding single-cell RNA-seq studies^{2,4} which also do not give confidence intervals for the same reasons. These papers are published in prestigious journals of *Cell* and *JCI Insight* (*Cell*. 2018;175(4):1014-1030 e1019; PMC6501221; *JCI Insight*. 2018;3(8); PMC5931117). In the revision, we include the denominator that applies to the percentages. This can be seen in the legend for Supplementary Figure S1 (now Supplementary Figure 3). The figure legend for Suppl. Figure 3 now includes this text: "Specifically, for the before anti-PD-1 treatment melanoma samples, the denominators (total number of CD45⁺ cells) for the R (responsive) and the NR (non-responsive) groups are 1191 and 2604, respectively; for the after anti-PD-1 treatment melanoma samples, the denominators for the R and the NR groups are 1524 and 6334, respectively; for the after anti-CTLA4 plus anti-PD-1 treatment melanoma samples, the denominators for the R and the NR groups are 1315 and 1190, respectively."

7. Please spell out what IPA analysis is when first presented.

Reply: We have done this as suggested.

8. Throughout the paper (e.g. p 5, bottom, and Figure S2), please indicate whether a p-value is a raw p—value or an adjusted p-value. If it is a raw p-value, and an adjustment is needed to properly interpret the p-value, please provide it as well.

Reply: The p values are adjusted by using Bonferroni correction. We added this clarification in the paragraph titled 'Gene expression heterogeneity in macrophage populations: implications for ICT resistance' in the revised manuscript on Page 6 as below. Original Figure S2 has been shifted to Figure S5 and the legend has been revised as suggested, too..

“IPA (Ingenuity Pathway Analysis) confirmed that inflammatory markers were significantly overexpressed in cluster 6 versus other macrophages (adjusted P = 3.93E-10, activation Z score = 2.01, Supplementary Figure 5 in Supplementary Methods; P values throughout this paper are adjusted by using Bonferroni correction).”

9. Something seems deeply flawed statistically in the very long eSpreadsheets which generate adjusted p-values that are so strikingly low (e.g. 3 and one-half pages of genes whose adjusted p-values are less than E-50.) It is unclear that such long lists produce any clear information. If they do, perhaps some greater explanation of them is needed.

Reply: In this study, the cell subtype-specific genes were identified using the FindAllMarkers function of Seurat, which has been a widely used software package for single cell RNA-seq analysis since its publication in Nature Biotechnology in 2015⁵. It has been used and cited in at least 143 publications (through the beginning of May, 2020) according to PubMed (<https://pubmed.ncbi.nlm.nih.gov/?term=Seurat>).

We have found that previous scRNA-seq studies based on Seurat analysis also generated adjusted p-values that are low, even for studies with much smaller numbers of cells than ours. For example, as was reported in a single cell RNA-seq of atherosclerosis³ (*Circ Res.* 2018;122(12):1661-1674. PMID: 29545365), which only involved 1226 single cells (compared to 16291 single cells in our study), their identified cell subtype-specific genes also showed very low p values as can be seen in their long list of results (Online Table 1 in the Supplemental Material of that study, from Page 17 to Page 122; Online Table 3 in the Supplemental Material of that study, from Page 125 to Page 224). Almost all the cell clusters gave very low p values; for example, in their study, many immune cell clusters had more than 3 pages of genes whose adjusted p-values are less than E-20. The p values of the most significant genes for CD8+ T cells, MoDC/DC, ResLike macrophages, inflammatory macrophages, B cells, monocytes, mixed T cells, neutrophils are 2.89E-170, 6.16E-111, 1.54E-115, 2.41E-104, 2.20E-169, 9.84E-84, 6.46E-106, and 3.72E-167, respectively.

In a technical post by the authors of Seurat (<https://github.com/satijalab/seurat/issues/205>), it was recognized that in the clustering analysis, it is normal to see strikingly low marker p values, many even returned as 0. The authors explained this as follows: “it's possible that for large datasets with strong markers that you would see some p-values of 0. It's not so much a Seurat specific feature, rather the smallest number that R can represent is 2.225074e-308. So any values less than that are just treated as 0.” Therefore, our results are expected given the large datasets and the strong cluster specific markers we identified.

We agree that such long lists may not yield clear information. That is the reason why we put them in the Supplementary files. They are just supplementary information to justify the existence of these cell subtypes based on their special gene expression patterns.

Reviewer #2 (Remarks to the Author):

The study by Xiong et al. focused on one public single-cell RNA-seq dataset of ICT-treated melanoma patients, and reanalyzed the immune cell populations in two groups of patients with different ICT outcomes. The authors compared the immune cell clusters between responders and non-responders, and discovered specific subsets of macrophages, $\gamma\delta$ T cells and B cells that were potentially correlated with ICT resistance in melanoma. Although the authors claimed to have revealed an immune signature that could predict ICT outcomes, this study in the end were highly descriptive, and the major discovers were all based on very limited analyses. In other words, the reliability of the conclusions of this study can be improved, and further analyses and experiments are needed to support the biological findings.

Major concerns,

1. It is difficult to know how robust the clustering result is, especially for the heterogeneity of macrophages, since there is little to no direct testing of the predicted key functionally, mechanistically or clinically relevant features of different subpopulations.

Reply: As stated in detail in response to point 2 and 3 below, we performed the statistical tests (Wilcoxon tests) at the patient level for each of the 23 immune cell clusters based on two groups containing 17 samples for the responsive group and 31 samples for the non-responsive group. The new analytical results (Supplementary Figure 2) are consistent with our original results and confirm the robustness of the initial clustering results. The biological meanings of the macrophages we identified, especially for TREM2^{hi} macrophages, are now discussed in greater detail. Please see our response to point 4 below. All these revisions have been included in the revised version.

2. The authors compared the percentages of each cell clusters in the integrative clustering result to differentiate cell frequencies between responders and non-responders, and drew the major conclusions of this study. However, this dataset involves 48 samples and each group contains enough number of samples to perform statistical analysis, and the authors should perform statistical tests to increase the credibility of such analyses.

Reply: Thank you for this important point. We have performed statistical tests (Wilcoxon tests) at the patient level for each of the 23 immune cell clusters based on the two groups (17 samples for responsive group and 31 samples for non-responsive group). These new results (Supplementary Figure 2) are very similar to the results that compared the percentages of each cell cluster in the integrative clustering to differentiate cell frequencies between responders and non-responders. This statistical analysis therefore supports the results from our original analyses. We have added the above on Page 5 of the revised manuscript and incorporated the corresponding figures of 23 clusters in Supplementary Figure 2.

3. Similarly, it is not appropriate to only utilize the overall fold differences of percentages of cell clusters in two patient groups to conclude the enrichment or deficiency of specific cell clusters in the resistant group (Fig 2B), since fold differences of cell frequencies are highly affected by the size of the cell frequencies of pre-treatment. Thus, statistical analyses should be performed at the patient level.

Reply: Thank you for this important comment. We have performed statistical analyses at the patient level. Please see Supplementary Figure 2 for the details. Clusters of C12, C21 and C22's percentages differ significantly at the patient level between the non-responsive and responsive patients. The results from the patient level analyses are consistent with the cell level analyses. We have added this new justification on Page 5 of the revised manuscript as follows: "Our approach is the same approach used in the previous scRNA-seq study² (*Cell*. 2018;175(4):1014-1030 e1019; PMC6501221.) of the effects of the immunotherapy on changing the percentages of different immune cell subpopulations. They compared the percentage of cells in individual clusters by different conditions of control, anti-PD-1, anti-CTLA-4, and anti-PD-1/anti-CTLA-4. In this way, they identified several significant immune cell subclusters that could be associated with the variation in the efficacy of cancer immunotherapy. In addition, we performed statistical tests (Wilcoxon tests) at the patient level for each of the 23 immune cell clusters based on the two groups (17 samples for responsive group and 31 samples for non-responsive group). The results (Supplementary Figure 2) are very similar to the results that compared the percentages of each cell cluster in the integrative clustering to differentiate cell frequencies between responders and non-responders. These new results support those from our original analyses."

4. The authors discovered that TREM2hi macrophages were enriched in ICT resistant group and assumed that this subset were functionally proximal to M2 polarization macrophages, which could block the anti-tumor activities of ICT. However, the single-cell studies by Azizi et al. (*Cell*, 174, 1293-1308 e36, 2018) and Wagner et al. (*Cell* 177, 1330-1345 e18, 2019) reported that M1 and M2 signatures are positively correlated in myeloid populations. The authors should check the expression of M1 markers in TREM2hi population. In addition, TREM2hi macrophages highly expressed complement system genes and enriched complement activation pathway, the function of which in anti-tumor immunity seems controversial (*Ann Transl Med*, 4(14): 265. 2016; *J Clin Invest*. 2017 Mar 1; 127(3): 780–789.2017). Thus, the authors should experimentally confirm the function of TREM2hi macrophages.

Reply: We checked the expression of M1 markers in the TREM2hi macrophages (Supplementary Figure 11) based on the M1 macrophage signature genes listed in the articles suggested by the reviewer. It was found that the expression of M1 signature genes was neither strong nor consistent. iNOS (NOS2), the most characteristic and canonical M1 macrophage marker, was not expressed in the TREM2hi cell population.^{2,6-8} These findings suggest that TREM2hi cells are functionally more proximal to M2 polarization macrophages. We have added these points to our discussion on Page 15.

Although the role of the complement system is not completely understood, as summarized by the paper the reviewer suggested,⁹ there are different mechanisms by which complement activation in the tumor microenvironment can enhance tumor growth, such as altering the immune profile of tumor-infiltrating leukocytes, increasing cancer cell proliferation, and suppressing CD8+ TIL function. It has also been found that complement effectors such as C1q,

C3a and C5a, etc., have been associated with inhibition of antitumor T-cell responses through the recruitment and/or activation of immunosuppressive cell subpopulations such as MDSCs (myeloid-derived suppressor cells), Tregs or M2 tumor-associated macrophages (TAMs).⁹ The rationale for inhibiting the complement system for new therapeutic combinations that aim to enhance the antitumor efficacy of anti-PD-1/PD-L1 checkpoint inhibitors has been proposed based on supporting evidence that complement blocks many of the effector routes associated with cancer immunity cycle.¹⁰ (*Front Immunol.* 2019;10:774; PMC6473060.) Our study results were in line with these new ideas and suggested that the TREM2hi macrophage population which has an activated complement system could be another source or consequence of complement activation contributing to the blockade of cancer immunity. Discussion of these points has been added into the revised manuscript on Page 14. In the future, we intend to perform functional assays on the TREMhi macrophages, although they are beyond the scope of this paper given that such studies are very lengthy.

5. The biological meaning of the enrichment of one specific subset of $\gamma\delta$ T cells in non-responder patients should be explored. Although the authors discovered that this subset of cells had reduced ligand-receptor binding capacity, IFN α and IFN β signaling, IFN- γ response, and immunoregulatory interactions, it is not enough to deduce their roles in mediating ICT resistance.

Reply: We agree with these comments. However, experiments to explore the biological meaning of the enrichment of the specific subset of $\gamma\delta$ T cells (Tgd_c21) in non-responder patients are beyond the scope of this manuscript. Interestingly, a previous study showed that the positive outcome of PD-1 blockade on treating leukemia may be because it induces significant upregulation of the potent pro-inflammatory and anti-tumor cytokine IFN- γ of certain types of $\gamma\delta$ T cells¹¹ (*Oncoimmunology.* 2019;8(3):1550618; PMC6350692). Complementing their study, we showed that the failure of immunotherapy in treating melanoma may be associated with some types of previously unrecognized $\gamma\delta$ T cells (e.g., Tgd_c21). The pathway analysis showed that this subset of $\gamma\delta$ T cells - Tgd_c21 had decreased activity of the anti-tumor IFN- γ pathway in the non-responders than the responders subjected to the immunotherapy (Figure 3C). Therefore, a key element may be the IFN- γ pathway activity, whose reduction in some $\gamma\delta$ T cell subsets such as Tgd_c21 in ICT non-responders may contribute to ICT resistance. We have added these discussion points to the revised manuscript on Page 15-16.

6. The authors identified a gene signature that could predict ICT outcomes from bulk data, and claimed that this signature was enriched for the characteristic genes of the three significant single-cell clusters - TREM2hi macrophages, Tgd_c21 $\gamma\delta$ T cells and B_c22 B cells. What is the predictive capability of the gene signature of these three cell clusters directly identified from single-cell data? And what's the difference between this two sets of signatures derived from different data resources?

Reply: As suggested, we used the gene signature representing the three component cell clusters (TREM2^{hi} macrophages, Tgd_c21 $\gamma\delta$ T cells and B_c22 B cells) identified from the scRNAseq data (Figure 2, Figure 3, Supplementary Figure 7). This 150-gene signature is composed of three sets of top 50 genes most significantly over-expressed in one of the three cell clusters (Supplementary Materials: the top ranked 50 genes for TREM2^{hi} macrophages in Spreadsheet 2, the top ranked 50 genes for Tgd_c21 $\gamma\delta$ T cells in Spreadsheet 7 and the top

ranked 50 genes for B_c22 B cells in in Spreadsheet 8). This gene signature was called scR.Immune and used for ICT outcome prediction. The scR.Immune signature had a somewhat lower predictive capability compared with the ImmuneCells.Sig signature derived from both scRNAseq and bulk gene expression datasets. As can be seen from Supplementary Figure 12, the AUC values from scR.Immune were 0.92, 0.90, 0.84 and 0.78 for the datasets of GSE78220, GSE91061, PRJEB23709 and MGSP, respectively, which are lower than the AUC values given by the ImmuneCells.Sig signature (0.98, 0.97, 0.92 and 0.88 for the four datasets, respectively). The difference in predictability between these two sets of signatures is likely due to the complex cellular composition of tumors. Because the four datasets used for AUC calculations are all bulk gene expression data, the corresponding expression levels of genes represented a mix of expression from all kinds of cells embedded in the tumor samples. So using scRNAseq data derived signature alone such as the scR.Immune signature may not predict ICT outcome better than using the ImmuneCells.Sig signature derived from both scRNAseq and bulk gene expression datasets. However, the ImmuneCells.Sig signature is enriched for the signature genes from the TREM2hi macrophages, Tgd_c21 $\gamma\delta$ T cells and B_c22 B cells, suggesting the involvement of these immune cell subpopulations in determining the ICT responsiveness. We have added these points to the discussion on Page 16-17.

Other points,

1. The result showed in barplot of Fig 2A could be improved with boxplot showing the variations of patient level.

Reply: We have included a boxplot showing variations of the patient level in Supplementary Figure 2. Thank you.

2. The authors should confirm the specificity of the 40-gene set identified to characterize TREM2hi macrophages to ensure the reliability of the confirmation analysis in bulk data.

Reply: We compared the GSVA scores of the 40-gene set identified across macrophages and verified the specificity of this gene set to characterize the TREM2hi macrophages as shown in the violin plot in Supplementary Figure 7D. The activity of this gene set is significantly higher in the TREM2hi macrophages compared to the other macrophages. We have included these details in the revised manuscript on Page 8.

3. It is better to move Fig 3A and Fig 4 to supplementary figures.

Reply: We have moved these as suggested.

4. IDO1 was reported to be one of the major markers of IDO1hi populations of macrophages, however, the violin plot in Fig 3C showed that this gene was hardly expressed in IDO1hi subset. Maybe the authors could rename this subpopulation with other specific genes.

Reply: Thank you for this suggestion. Based on analysis of its specific genes, we have renamed this population as 'Inflammatory M ϕ '.

5. It is better to show the enrichment of predictive signature in markers of three cell clusters with other forms of figure resembling GSEA analysis, rather than to show the expressions of selected markers in each cluster (Fig 5B).

Reply: As suggested, we have replaced Figure 5B with three new figures of GSEA results showing that the predictive signature - ImmuneCells.Sig was enriched for the characteristic genes of TREM2^{hi} M ϕ , Tgd_c21, B_c22 immune cell subpopulations. Figure 5 in the original manuscript is now Figure 4 in this revision, so these new figures are now 4B, 4C and 4D. The original figure 5B has been removed, and the original Fig 5C and 5D are now Fig 4E and 4F.

Reviewer #3 (Remarks to the Author):

In summary, this work reports the identification of a new single-cell-based RNA signature that could be predictive of clinical outcome for patients treated with immune checkpoint therapies (ICT) for melanoma. The authors have worked on the dataset from a previously published work by Sade-Feldman et al (reference 17) where 48 biopsies from patients treated with ICT were analyzed by scRNAseq. Rather than focusing on CD8 lymphocyte as in the previous study, they study other immune cell compartment, such as myeloid population, rare $\gamma\delta$ T cells and B cells. 23 clusters of immune cells were individualized based on the sc-RNAseq and 3 of these clusters showed more than 6 fold differences between NR and R patients. Clusters 12 and 21 were more represented in NR patients and corresponded respectively to a subset of macrophages and to proliferating $\gamma\delta$ T cells, whereas cluster 22, a B cell cluster was less represented in NR samples. This was done initially on all samples, then analyzed separately with regards to pre or post ICT as well as to the type of ICT regimens, where the same tendencies were observed (although on extremely small samples of patients).

More detailed characterization of cluster 12 showed that it was related to a TREM2^{high} population of macrophages expressing M2 polarization genes as well as several complement system genes. A TREM2 macrophage signature of 40 genes was designed and found to be associated with NR when interrogating publicly available patients' RNA signatures.

A neosignature named ImmuneCells.Sig, containing over 90 genes integrating scRNA variations and bulk RNA data from a previous analysis (GSE78220) was defined. Unsurprisingly it was well correlated to the outcome when tested on the patients from the GSE78220 since this data was used to create the signature. Finally, the authors find that this ImmuneCells.Sig compares favorably to other signatures that are associated with ICT response to date.

Comments & questions:

Patients' material:

1. Drawing general conclusions based on a mix of pre- and post treatment biopsies and mixing various treatment regimens does not make a lot of sense when one aims at identifying predictive biomarkers because the predictive signature can be used in the clinic only when

analyzing pre-treatment biopsies (or early on-treatment biopsies). We must thus acknowledge the efforts done by the authors in performing differential analyses on pre-treatment samples, then on post-treatment samples or on samples from patients with different types of treatments. But then, we see that we only have a very small group of patients in the pre-ICT situation: 8 non-responders versus 4 responders before anti PD1. All the other biopsies that could be analyzed were taken post-ICT. Of course, the immune infiltrate of these post-ICT samples could have been modified by the treatment. This does not mean that it is not interesting to study but it addresses a question distinct from the identification of predictive baseline markers. How do the authors explain that data found on such a small number of pre-therapy samples could be useful to derive a pre-treatment signature?

Reply: Although the number of patients from which pre-treatment samples were available is not large, hundreds to thousands of single cells were analyzed from each of these samples to characterize the individual cell clusters. In this revision, we have included GSEA analyses showing that the predictive signature - ImmuneCells.Sig was enriched for the characteristic genes of TREM2^{hi} M ϕ , Tgd_c21, B_c22 immune cell subpopulations (Figure 4). Two independent scRNA-seq datasets have been analyzed and the results support our findings (please also see our responses to your points 6 and 7). We also used the three bulk RNA-seq datasets (including a new analyzed cohort of 103 pretreatment melanoma samples) to validate the utility of the derived signature in pre-treatment melanoma samples in terms of predicting the response to ICT treatment. In addition, we also compared the performance of the ImmuneCells.Sig to the other 12 outstanding ICT response signatures in the field across all the four bulk RNAseq datasets (GSE78220, GSE91061, PRJEB23709, and MGSP datasets) and found it is superior to the other signatures. Only 1 of 28 samples in GSE78220 was early-on-treatment sample, the rest 27 are all pre-treatment samples. All of the samples in the other three datasets are the pretreatment samples, i.e., the 51 pre-treatment samples from GSE91061, the 73 pre-treatment samples from PRJEB23709, and the 103 pre-treatment samples from MGSP. The converging evidence indicates that the findings from the single cell RNA-seq datasets combined with the bulk RNA-seq data can be used to derive a pre-treatment signature.

2. Sup fig 1A is very misleading and cannot be used to make the point the authors want to make. Indeed, GSE79691 data is not about 10 patients but reports the results from 10 biopsies taken post-mortem from the same patient. This post-therapy data is thus not predictive of response but shows the heterogeneity of the immune response to ICT in various metastases from the same patient.

Reply: To avoid confusion, we have deleted the Sup fig 1A that was in the original version. The GSE79691 dataset was only used one time as in the Sup fig 3B in the original version; therefore, we have also deleted the original Sup fig 3B. We do not use the GSE79691 dataset in this revised manuscript.

3. For the data from PRJEB23709: what are the 27NR and 46 R from the Gide et al manuscript : anti PD1 or combination? Pre and pot therapy?

Reply: We used the scRNA-seq data (PRJEB23709) of the 73 pre-treatment tumors from the study by Gide et al. Among these 73 samples, 41 are from melanoma patients subjected to anti-PD-1 therapy and consist of 19 non-responders and 22 responders; the other 32 samples are

from melanoma patients subjected to combined therapy with anti-PD-1 and anti-CTLA-4; of these, 8 were non-responders and 24 were responders. We have added these details to the Method section on Page 18 in the revised manuscript.

4. Could the authors show cohorts data that are from pre-ICT biopsies and using only anti PD1 or only the combination of anti PD1 and anti CTLA4? This would help to know if the new signature could have a real application in the clinic. Who to treat based on pre-therapy biopsy? Who to treat with the toxic combination regimen anti CTLA4 + anti PD1 versus the less toxic anti PD1?

Reply: Among the four bulk RNAseq datasets, only the PRJEB23709 dataset¹² (Gide et al. *Cancer Cell*. 2019;35(2):238-255 e236) had pre-ICT biopsies for melanoma patients treated with either anti-PD-1 or the combination of anti PD-1 and anti-CTLA-4 drugs. Accordingly, we had used the bulk RNAseq data of the 73 pre-treatment tumors from the study by Gide et al. (PRJEB23709) which included 41 patients which received only anti-PD-1 treatment (19 non-responders vs 22 responders), and 32 patients which received the combination of anti-PD-1 and anti-CTLA-4 treatment (8 non-responders vs 24 responders). We split the PRJEB23709 dataset into PRJEB23709_Pre_anti-PD-1 and PRJEB23709_Pre_anti-Combo according to the treatment scheme (anti-PD-1 or combination of anti PD-1 and anti-CTLA-4). In each dataset, we tested the performance of ImmuneCells.Sig. It was found that ImmuneCells.Sig can accurately distinguish responders from non-responders in both Pre_anti-PD-1 and Pre_Combo subgroups. For PRJEB23709_Pre_anti-PD-1 subset, the performance of ImmuneCells.Sig is as follows: AUC = 0.88 (95% CI, 0.83 to 0.94), sensitivity = 86% (95% CI, 68% to 96%), and specificity = 79% (95% CI, 58% to 92%) (Supplementary Figure 10A). For PRJEB23709_Pre_Combo subset, the performance of ImmuneCells.Sig is as follows: AUC = 0.93 (95% CI, 0.86 to 0.99), sensitivity = 88% (95% CI, 71% to 97%), and specificity = 88% (95% CI, 53% to 99%) (Supplementary Figure 10B).

Using the 'cancerclass' R package, we can calculate the z-score in each pre-therapy biopsy based on the expression values of the ImmuneCells.Sig genes to predict who are more likely to respond to anti-PD-1 or anti-PD-1 plus anti-CTLA-4 combo therapy. For example, in the model built from Pre-anti-PD-1 dataset of PRJEB23709_Pre_anti-PD-1, the threshold z-score of 0.19 yielded sensitivity of 91% for responders. In the model built from Pre-Combo dataset of PRJEB23709_Pre_Combo, the threshold z-score of 0.1 yielded sensitivity of 91% for responders. Therefore, if we test a pre-therapy melanoma sample, the corresponding patient may not respond to either anti-PD1 treatment or anti-PD-1 plus anti-CTLA-4 combo treatment if the z-score < 0.1, but may respond to the more toxic combo treatment if z-score is within the range of [0.1, 0.19], and may respond to the less toxic anti-PD-1 treatment alone if the z-score > 0.19. Therefore, prediction of the outcomes of different therapy regimen is possible based on the application of ImmuneCells.Sig. We added the above in the Results section of the revised manuscript on Page 12-13.

Concerning the definition and interpretation of the scRNA signatures results:

1. With regards to the previous published study by Sade-Feldman et al: the authors should mention that the increased B cells and macrophages in NR patients is not a new data as it was

already shown in the initial paper. Figure 1 and 2 here do not bring two different message and could be presented in one figure.

Reply: Thank you for the helpful comment. In the discussion, we have now added the statement that the increased B cells and macrophages in NR patients had already been reported in the initial paper. We have combined Figures 1 and 2 into one figure (Figure 1 in this revision). We also moved the original Figure 1C to Supplementary Figure 1A in this revision.

2. How do the authors explain that the CD8⁺ T cells are found less represented in their analysis in responding patients whereas it was found increased in the initial paper? This is an obvious discrepancy that is not mentioned in the manuscript.

Reply: Their initial paper divided CD8⁺ T cells into four groups, G6-Exhausted CD8⁺ T cells; G8-Cytotoxicity (Lymphocytes); G9-Exhausted/HS CD8⁺ T-cells; G10-Memory T-cells. Among these, two groups (G6 and G10) have significant percentage changes between responders and non-responders. Actually, the G6 group of exhausted CD8⁺ T cells were less represented in responders while the CD8⁺ memory T cells from the G10 groups were enriched in responders. Our analyses in the current paper led to higher clustering levels and divided CD8⁺ T cells into 7 groups (clusters 1, 4, 5, 7, 10, 11 and 20). From this new analysis, responders had decreased percentages of CD8⁺ T cells from clusters 1, 10, 11, and 20 defined in our paper (Supplementary Table 1), which matches the direction of the G6 group in their initial paper. On the other hand, the responders had increased percentages of CD8⁺ T cells from clusters 4, 5, and 7 defined in our paper (Supplementary Table 1), which matches the direction of the G10 group in their initial paper. Therefore, our results do not contradict the prior paper, but the higher degree of clustering provided here better delineated changes to specific subsets of CD8⁺ T cells.

3. There is a concern about the definition of $\gamma\delta$ Tcells based on the combination CD3+CD4-CD8- only that could also correspond to NK cells that are in cluster 15, quite close to clusters 8 and 21 and can also be CD4-CD8-. Is the unique constant region of the TCR δ chain (TRDC) over-represented in the cluster 8 and 21? Does this population express the receptors TCRV δ 1 or TCRV δ 2? Altogether, the definition of what is here considered to be $\gamma\delta$ Tcells should be validated by the expression of additional $\gamma\delta$ Tcells signatures published in ImmGen and with published gene signatures of $\gamma\delta$ T cells (for example PNAS June 11, 2019 116 (24) 11906-11915).

Reply: The NK cells in cluster 15 expressed the NK cell markers NCR1 and NCAM1, which were not expressed in $\gamma\delta$ T cells in clusters 8 and 21 (Supplementary Figure 1). The NK cells (cluster 15) also do not express CD3 markers, whereas CD3 markers were expressed in the adjacent clusters (8 and 21) that were characterized as $\gamma\delta$ T cells based on the combination CD3+CD4-CD8-. In addition, as suggested, we have now validated our defined $\gamma\delta$ T lymphocytes by the expression of the published gene expression signatures of $\gamma\delta$ T cells¹³ (*Proc Natl Acad Sci U S A.* 2019;116(24):11906-11915; PMC6576116), which requires scoring the following two gene sets: the positive gene set (*CD3D, CD3E, TRDC, TRGC1, TRGC2*), and the negative gene set (*CD8A, CD8B*) for each single cell. Specifically, following this published approach¹³ to identify $\gamma\delta$ T lymphocytes exhaustively and without NK and T cell CD8 false-positives, we utilized the established $\gamma\delta$ signature that combines the above two gene sets that

was scored for each single cell and visualized in the t-SNE by Single-Cell Signature Explorer.¹⁴ As shown in Supplementary Figure 1B in this revision, the $\gamma\delta$ signature scores were highest for clusters 8 and 21, and were much lower in the other clusters. These data further support our assignment of $\gamma\delta$ T lymphocytes to clusters 8 and 21. We have added these details in the revised manuscript on Page 4-5 of results section and Supplementary Figure 1.

4. How does the present B cell signature compare with data on B cells recently published by several teams in the context of ICT? This was not discussed in the text.

Reply: Thank you for this important point. We have done additional analysis and have added the following discussion to the revised manuscript on Page 16. "We compared the present B cell subpopulation signature (B_c22, derived from eSpreadsheet 8 based on the cutoff P value of 0.05) with the other B cell signature recently published in the context of ICT by Helmink et al¹⁵, and found that several genes were shared by both signatures, including TCL1A, ITIH5, LAX1, KCNA3, CD79A, AREG, GBP1, ATP8A, and IGLL5. Both our signature and the signature of Helmink et al. characterized B cell populations that were significantly enriched in the ICT responders versus non-responders. However, the B cells associated with these two signatures were not identical. This is because our B_c22 (single cell cluster 22) signature was developed based on the scRNA-seq data of melanoma samples whose corresponding B cells were a subset of B cells that were highly enriched in the ICT responders versus the non-responders (> 9-fold, Figure 1 and Table S1). We also identified an additional three B cell subpopulations (clusters 13, 14, 17) (Figure 1 and Table S1). In contrast, the B cell signature used by Helmink et al. was derived from bulk RNA-seq data of renal cell carcinoma (RCC); thus their signature may represent a mix of B-cell subpopulations in RCC patients that responded to ICT. Therefore, it is logical for the two signatures to share several, but not all, genes."

5. The authors find that the B cell cluster 22 has reduced oncogenic pathways expression. How could this explain that they could contribute to the activation of the oncogenic signaling in the tumor environment? This seems as gratuitous suggestion with no evident logical explanation, and no mechanism explored.

Reply: Thank you for this point. We have revised the text as follows on Page 9: "The significant enrichment of B_c22 cells in ICT responders may therefore contribute to the attenuation of oncogenic signaling in the tumor microenvironment (TME), which could enhance the anti-tumor effect in response to ICT."

6. Additional datasets of scRNAseq are available, in particular, the one published in cell by Jerby-Arnon et al (<https://doi.org/10.1016/j.cell.2018.09.006>). Using this additional data should be done as it could strengthen the results.

Reply: Thank you for this important point. To validate our results that were based on the initial scRNAseq data, we downloaded and re-analyzed the scRNAseq dataset of melanoma and immunotherapy efficacy from the study by Jerby-Arnon et al.¹⁶ This scRNAseq dataset did not have $\gamma\delta$ T cells data available. Interestingly, the deeper clustering of the macrophages and B cells sequenced by this study¹⁶ showed the existence of similar macrophage and B cell subpopulations that resemble our identified TREM2^{hi} macrophages and B_c22 B cells (Supplementary Figure 8A, 8B). Specifically, the 'Mac_c1' macrophage subcluster

overexpressed the TREM2^{hi} macrophage marker genes (*TREM2*, *SPP1*, *RNASE1*, *MT1G*, *SEPP1*, *FOLR2*, *KLHDC8B*, *CCL18*, *MMP12*, *APOC2*) (Supplementary Figure 8C); the 'B_s1' B cell subcluster overexpressed the B_c22 B cells marker genes (*ABCA6*, *LEF1*, *FGR*, *IL2RA*, *ITGAX*, *IL7*) (Supplementary Figure 8D). More importantly, we validated the behavior of these two immune cell subpopulations in the context of the response to immunotherapy. We scored each cell based on its overall expression (OE) of the corresponding signature following the approach by Jerby-Arnon et al.¹⁶, i.e., scoring each Mac_c1 macrophage for its TREM2^{hi} macrophage signature and each B_s1 B cell for its B_c22 B cell signature, and compared these between the non-responder and control groups. In this new dataset, the Mac_c1 macrophage subset had significantly higher overall expression of the TREM2^{hi} macrophage signature in the immunotherapy non-responders than in the control samples (Supplementary Figure 8E). The B_s1 B cell subset had significantly lower overall expression of the B_c22 B cell signature in the immunotherapy non-responders than in the control samples (Supplementary Figure 8F). These results supported the findings in our initial scRNAseq dataset of the changes in TREM2^{hi} macrophages and B_c22 B cells in response to immunotherapy. We have added the above text on page 9-10 of the revised manuscript under the subheading "Validation of the findings in melanoma using other scRNAseq datasets of melanoma and basal cell carcinoma".

7. The presence of certain immune cells, especially macrophages and gamma-delta-T cells could very well depend on the type of cancer or the type of organs involved. This origins of the metastases are thus important to mention and to analyze with regards to the type of infiltrate. It would also be interesting to look at various tumor types and not only melanoma patients to see if the results can be generalized to other types of cancers treated with ICT.

Reply: We have now also analyzed a single cell RNA-seq dataset of basal cell carcinoma (BCC) patients before and after anti-PD-1 therapy.¹⁷ We found that the results of our study can be generalized to BCC treated with ICT. Although this BCC scRNAseq dataset did not sequence the $\gamma\delta$ T cells, the results for macrophages and B cells in this BCC dataset are similar to our findings for the melanoma dataset. Based on the BCC scRNAseq dataset, first, we did general clustering analyses and identified the overall macrophages and B cell populations (Supplementary Figure 9A). Then we performed finer clustering and identified the macrophage and B cell subpopulations from the BCC tumors that are similar to the TREM2^{hi} macrophages and B_c22 B cells in the initial melanoma samples (Supplementary Figure 9B, C, D, E). In the BCC dataset, the 'Mac_s2' macrophage subcluster overexpressed the TREM2^{hi} macrophage marker genes (*TREM2*, *FOLR2*, *MMP12*, *C1QA*, *C1QB*, *C1QC*) (Supplementary Figure 9D); the 'B_sc2' B cell subcluster overexpressed B_c22 B cells marker genes (*TRAC*, *IL2RA*, *ITGB1*, *ZBTB32*, *TRAF1*, *CCND2*) (Supplementary Figure 9E). As before, we validated the overall expression changes of the TREM2^{hi} macrophage signature of the Mac_s2 macrophages and the B_c22 signature of the B_sc2 B cells in response to anti-PD-1 immunotherapy in this BCC dataset.¹⁷ Specifically, the Mac_s2 macrophage subset had significantly decreased overall expression of the TREM2^{hi} macrophage signature in the responsive BCC tumors after the anti-PD-1 therapy when compared to the pretreatment BCC samples (Supplementary Figure 9F). The B_sc2 B cell subset had significantly higher overall expression of the B_c22 signature in the responsive BCC tumors after the anti-PD-1 therapy than in the pretreatment BCC samples (Supplementary Figure 9G). These findings suggest that the immune cell subpopulations that we had identified as associated with the outcomes of cancer immunotherapy for melanoma also exist in BCC, and that the characteristic gene expression signatures may be altered similarly in

melanoma and in BCC in the context of response to immunotherapy. We have added these additional details on page 9-10 under the subheading “Validation of the findings in melanoma: comparing scRNAseq datasets of melanoma and basal cell carcinoma”.

8. TREM2^{hi} macrophage were already reported in breast cancer single cell studies (Azizi et al, Cell, 2018) to be a branch of recruited or resident M2 type macrophages expressing several genes in common with present study: SSP1, C1Q, CCL18, MACRO. This is thus not a new data.

Reply: We agree with this point. In the Discussion in the original submission, we had stated that “Many M2 polarization genes, some of which are known to be tumor-promoting, were also overexpressed in TREM2^{hi} macrophages.” However, TREM2^{hi} macrophages had not been linked to cancer immunotherapy response before. So that aspect of our data is novel and valuable to clinical practice in cancer immunotherapy. We included the reviewer’s comment and our above response into the Discussion of the revised manuscript on Page 14-15.

In Conclusion: This work reports descriptive data derived from a secondary analysis of a scRNAseq dataset and leading to a new predictive signature for ICT efficacy that seems more robust than other RNA signatures in the same context. The definition of certain clusters is suboptimal and no mechanistic data are presented. Several hypotheses and statements are not substantiated by clear data. The signature seems interesting although its use remains in the research domain and not yet applicable in the clinic.

Reply: The definition of certain clusters has been improved according to the reviewer’s suggestions as can be seen in the response to Point 3 in the section “Concerning the definition and interpretation of the scRNA signatures results”. Our hypotheses and statements have been better substantiated as can be seen in our responses to the reviewer’s comments and more validation using three new datasets. Specifically, in the validation phase, we utilized two additional new scRNAseq datasets (Jerby-Arnon et al. *Cell*. 2018;175(4):984-997 e924; PMC6410377; Yost KE et al. *Nat Med*. 2019;25(8):1251-1259; PMC6689255) and one large cohort of pre-treatment melanoma samples (N = 103) (Liu et al. *Nat Med*. 2019;25(12):1916-1927; PMC6898788) in this new revision. The validation using the new scRNAseq datasets can be seen in the response to the reviewer’s Point 6 and 7 in the section “Concerning the definition and interpretation of the scRNA signatures results”. The validation based on the new large cohort of pre-treatment melanoma samples was presented in Figures 4G, 5D, 5H. The details are shown in the following paragraph, which have been included in this new revised manuscript on Page 11-13. The potential clinical application of the ImmuneCells.Sig can be seen in our response to Point 4 in the section “Comments & questions”.

“For further validation, we downloaded and analyzed the gene expression profile of a big cohort of melanoma patients who were treated by the anti-PD-1 immunotherapy, from which a large number of pretreatment melanoma samples from 103 patients with distinct response to ICT (46 responders vs 57 non-responders) had been subjected to RNA-seq¹⁸ (*Nat Med*. 2019;25(12):1916-1927; PMC6898788). In this large dataset that was called ‘melanoma genome sequencing project’ (MGSP, https://www.ncbi.nlm.nih.gov/projects/gap/cgi-bin/study.cgi?study_id=phs000452.v3.p1), the predictive value of ImmuneCells.Sig was still high. Specifically, it differentiated non-responders from responders with an AUC of 0.88 (95% CI, 0.84 to 0.91), sensitivity of 86% (95% CI, 68% to 96%), and specificity of 79% (95% CI, 67% to 88%) (Figure 4G).”

“The results show that the ImmuneCells.Sig that we developed was consistently the best signature for predicting response to immunotherapy across all four datasets (Figure 5). In comparison, the previously established IMPRES signature performed second best in the PRJEB23709 dataset (Figure 5C and 5G), fourth in the MGSP dataset (Figure 5D and 5H), and fifth in the GSE78220 and GSE91061 datasets (Figure 5A, 5B, 5E and 5F).“

References

- 1 Budczies, J. *et al.* cancerclass: An R Package for development and validation of diagnostic tests from high-dimensional molecular data. *J Stat Software* **59**, 1-19 (2014).
- 2 Gubin, M. M. *et al.* High-Dimensional Analysis Delineates Myeloid and Lymphoid Compartment Remodeling during Successful Immune-Checkpoint Cancer Therapy. *Cell* **175**, 1014-1030 e1019 (2018).
- 3 Cochain, C. *et al.* Single-Cell RNA-Seq Reveals the Transcriptional Landscape and Heterogeneity of Aortic Macrophages in Murine Atherosclerosis. *Circ Res* **122**, 1661-1674 (2018).
- 4 Yan, Y. *et al.* CX3CR1 identifies PD-1 therapy-responsive CD8+ T cells that withstand chemotherapy during cancer chemoimmunotherapy. *JCI Insight* **3** (2018).
- 5 Satija, R., Farrell, J. A., Gennert, D., Schier, A. F. & Regev, A. Spatial reconstruction of single-cell gene expression data. *Nat Biotechnol* **33**, 495-502 (2015).
- 6 Mould, K. J., Jackson, N. D., Henson, P. M., Seibold, M. & Janssen, W. J. Single cell RNA sequencing identifies unique inflammatory airspace macrophage subsets. *JCI Insight* **4** (2019).
- 7 Azizi, E. *et al.* Single-Cell Map of Diverse Immune Phenotypes in the Breast Tumor Microenvironment. *Cell* **174**, 1293-1308 e1236 (2018).
- 8 Wagner, J. *et al.* A Single-Cell Atlas of the Tumor and Immune Ecosystem of Human Breast Cancer. *Cell* **177**, 1330-1345 e1318 (2019).
- 9 Afshar-Kharghan, V. The role of the complement system in cancer. *J Clin Invest* **127**, 780-789 (2017).
- 10 Pio, R., Ajona, D., Ortiz-Espinosa, S., Mantovani, A. & Lambris, J. D. Complementing the Cancer-Immunity Cycle. *Front Immunol* **10**, 774 (2019).
- 11 Hoeres, T., Holzmann, E., Smetak, M., Birkmann, J. & Wilhelm, M. PD-1 signaling modulates interferon-gamma production by Gamma Delta (gammadelta) T-Cells in response to leukemia. *Oncoimmunology* **8**, 1550618 (2019).
- 12 Comprehensive genomic characterization of squamous cell lung cancers. *Nature* **489**, 519-525 (2012).
- 13 Pizzolato, G. *et al.* Single-cell RNA sequencing unveils the shared and the distinct cytotoxic hallmarks of human TCRVdelta1 and TCRVdelta2 gammadelta T lymphocytes. *Proc Natl Acad Sci U S A* **116**, 11906-11915 (2019).
- 14 Pont, F., Tosolini, M. & Fournie, J. J. Single-Cell Signature Explorer for comprehensive visualization of single cell signatures across scRNA-seq datasets. *Nucleic Acids Res* **47**, e133 (2019).
- 15 Helmink, B. A. *et al.* B cells and tertiary lymphoid structures promote immunotherapy response. *Nature* **577**, 549-555 (2020).
- 16 Jerby-Arnon, L. *et al.* A Cancer Cell Program Promotes T Cell Exclusion and Resistance to Checkpoint Blockade. *Cell* **175**, 984-997 e924 (2018).
- 17 Yost, K. E. *et al.* Clonal replacement of tumor-specific T cells following PD-1 blockade. *Nat Med* **25**, 1251-1259 (2019).

- 18 Liu, D. *et al.* Integrative molecular and clinical modeling of clinical outcomes to PD1 blockade in patients with metastatic melanoma. *Nat Med* **25**, 1916-1927 (2019).

REVIEWERS' COMMENTS:

Reviewer #1 (Remarks to the Author):

The authors have responded well and thoroughly to the issues that I raised.

Reviewer #2 (Remarks to the Author):

In the revised manuscript, the authors did additional analyses to answer my questions. Although most of the concerns have been resolved, the most important point of deciphering the biological meanings of the relevant cell types, including TREM2+ macrophages and a subset of $\gamma\delta$ T cells, which impact the efficacy of ICT treatment revealed in this study remain unsolved, and more efforts should have been put either through experimental strategies or through computational approaches such as unraveling their interacting cell types to strengthen the biological findings of their study. In addition, the clusters showing significance in newly added supplementary Figure 2 might be illustrated in main Figure 1, to substitute the panel c and d in original Figure 1, as the results showing patient level differences are more robust than those of the integrative analyses.

Reviewer #3 (Remarks to the Author):

The manuscript has been improved in this edited version. This signature opens new avenues of research but is probably not a biomarker that physicians are going to use in the clinic. From the scientific point of view, the work is interesting but remains descriptive and lacks mechanistic approaches.

We are grateful to the reviewers for their insightful and helpful comments. Please find below our detailed responses to the reviewers' comments. The revisions were made correspondingly in the manuscript or the supplementary files.

We are very grateful for your consideration.

Warm regards,

Ming You, MD, PhD
Joseph F. Heil, Jr. Professor in Molecular Oncogenesis
Associate Provost for Cancer Research
Director, Center of Disease Prevention Research
Professor of Pharmacology and Toxicology
Email: myou@mcw.edu
Phone: 414-955-2565

REVIEWERS' COMMENTS:

Reviewer #1 (Remarks to the Author):

The authors have responded well and thoroughly to the issues that I raised.

Response: Thanks.

Reviewer #2 (Remarks to the Author):

In the revised manuscript, the authors did additional analyses to answer my questions. Although most of the concerns have been resolved, the most important point of deciphering the biological meanings of the relevant cell types, including TREM2+ macrophages and a subset of $\gamma\delta$ T cells, which impact the efficacy of ICT treatment revealed in this study remain unsolved, and more efforts should have been put either through experimental strategies or through computational approaches such as unraveling their interacting cell types to strengthen the biological findings of their study. In addition, the clusters showing significance in newly added supplementary Figure 2 might be illustrated in main Figure 1, to substitute the panel c and d in original Figure 1, as the results showing patient level differences are more robust than those of the integrative analyses.

Response: Thank you for this advice. We discussed the limitation mentioned by the reviewer in the manuscript as follows: "A limitation of this study is that deciphering the biological meanings of the above relevant cell types that impact the efficacy of ICT treatment remains unsolved. Well-designed experimental strategies should be used to explore the hidden mechanisms to strengthen the biological findings of this study."

To address the comment on Figure 1, the results of the nine significant immune cell clusters of Supplementary Figure 2 (Clusters 6, 9, 12, 13, 14, 17, 19, 21, 22) have been compiled into a new figure and used to replace the old Fig.1c. We removed the old Fig.1c but kept Figure 1d for ease of comparison of the results between patient-level analysis and integrative analysis. We also made corresponding changes in the main manuscript to accommodate the Figure 1 changes. Thank you for this constructive suggestion.

Reviewer #3 (Remarks to the Author):

The manuscript has been improved in this edited version. This signature opens new avenues of research but is probably not a biomarker that physicians are going to use in the clinic. From the scientific point of view, the work is interesting but remains descriptive and lacks mechanistic approaches.

Response: This is similar to Reviewer #2's comment above. We discussed the limitation that can be seen in our response to Reviewer #2. Thanks.